# An integer GARCH model for a Poisson process with time-varying zero-inflation

Isuru Panduka Ratnayake[1]*, V. A. Samaranayake[2]

**1** Department of Biostatistics and Data Science, Kansas University Medical Center, Kansas City, KS, United States of America, **2** Department of Mathematics and Statistics, Missouri University of Science and Technology, Rolla, MO, United States of America

* rratnayake@kumc.edu

**Data Availability Statement:** All relevant data are within the manuscript and its Supporting Information files.

## Abstract

A serially dependent Poisson process with time-varying zero-inflation is proposed. Such formulations have the potential to model count data time series arising from phenomena such as infectious diseases that ebb and flow over time. The model assumes that the intensity of the Poisson process evolves according to a generalized autoregressive conditional heteroscedastic (GARCH) formulation and allows the zero-inflation parameter to vary over time and be governed by a deterministic function or by an exogenous variable. Both the expectation maximization (EM) and the maximum likelihood estimation (MLE) approaches are presented as possible estimation methods. A simulation study shows that both parameter estimation methods provide good estimates. Applications to two real-life data sets on infant deaths due to influenza show that the proposed integer-valued GARCH (INGARCH) model provides a better fit in general than existing zero-inflated INGARCH models. We also extended a non-linear INGARCH model to include zero-inflation and an exogenous input. This extended model performed as well as our proposed model with respect to some criteria, but not with respect to all.

## Introduction

The standard Poisson point process, which assumes statistical independence between observations, is not suitable for modelling time series of counts that display serial dependence. For example, counts of infectious disease occurrences can be considered as dependent on previous counts because of the infectious nature of the illness. One way to address this deficiency is to define a time series where the count at a given time is generated by a Poisson distribution whose intensity parameter is dependent on past counts and past intensities. Observe that for a discrete time process defined in such a manner with a linear dependent structure and observed at equally spaced time points, the intensity parameter at a given time is the mean count at that time conditional on the past information. Since many count data processes have an excess of zeros than what is possible by the underlying probability distribution, that is they exhibit zero-inflation, modelling serial dependence is not sufficient. Existing models for count data that accommodate serial dependence and zero-inflation, however, do not allow for the zero-inflation probability to vary over time cyclically or be driven by an exogenous variable. This

**Funding:** Enter: The author(s) received no specific funding for this work.

**Competing interests:** The authors have declared that no competing interests exist.

becomes a handicap when modelling count data time series, such as infectious disease or death counts that vary seasonally or with respect to an external factor that varies with time. To address this shortcoming, we propose the Time-Varying Zero-Inflated Poisson integer-valued GARCH model (TVZIP-INGARCH), which is based on a zero-inflated Poisson process whose intensity parameter satisfies the integer-valued GARCH (INGARCH) formulation introduced by Ferland, Latour, and Oraichi [1]. It is also a generalization of the model introduced by Zhu [2] which assumes constant zero-inflation. The proposed model can accommodate cases where the zero-inflation probability is driven by a deterministic function of time, such as a sinusoidal wave, or by exogenous variables. Before we introduce this model formerly, we will discuss some relevant INGARCH type count data time series models that have been proposed in the past. Note that we extended an existing non-linear INGARCH model without zero-inflation to include a zero-inflation component as well as an exogenous input variable. The performance of this new formulation and several existing INGARCH type models with constant zero-inflation were compared to that of the proposed model with respect to model fit on two real-life data sets.

Rydberg and Shephard [3] were the first to propose a count data time series model that account for serial dependence. In their model, the current conditional mean is a linear function of both the observed count and the conditional mean at the pervious time point. Similar models have been proposed by several other authors and these are discussed in Chapter 4 of the book by Kedem and Fokianos [4]. Heinen [5] generalized the lag one model of Rydberg and Shephard to include an arbitrary number of lags for both the past counts and past conditional means and named it the Autoregressive Conditional Poisson model with lags $p$ and $q$ (ACP $(p, q)$). The formulation of this model resembles that of the generalized autoregressive conditional heteroscedastic (GARCH) model of Bollerslev [6], but models the conditional mean rather than the conditional variance. Heinen derived the properties of his model only for the ACP (1, 1) case. The general case was investigated by Ghahramani and Thavaneswaran [7] who referred to the Heinen paper as the origin of the ACP model. Independently, Ferland et al. [1] proposed what was termed the Integer-valued GARCH (INGARCH) process, which is essentially the same as the ACP model of Heinen.

The INGARCH model of order $(p, q)$ is defined as follows:

$$X_t|\mathbb{F}_{t-1} \sim P(\lambda_t); \forall t \in \mathbb{Z}$$
$$\lambda_t = \alpha_0 + \sum_{i=1}^{p}\alpha_i X_{t-i} + \sum_{j=1}^{q}\beta_j \lambda_{t-j}, \tag{1}$$

where $\{X_t : t \in \mathbb{Z}\}$ is the count process, $\lambda_t$ defines the conditional mean of $X_t$ given the sigma-field $\mathbb{F}_{t-1}$ generated by $X_l$ and $\lambda_l$ values for $l < t$. The model is defined with the parameter constraints $\alpha_0 > 0, \alpha_i \geq 0, \beta_j \geq 0, i = 1, \ldots, p, j = 1, \ldots, q, p \geq 1, q \geq 0$, for positivity of $\lambda_t$ and

$0 \leq \sum_{i=1}^{p}\alpha_i + \sum_{j=1}^{q}\beta_j < 1$, for stationarity of $X_t$. In addition to presenting the model, Heinen [5]

also derived the stationarity conditions, covariance functions, and addressed the problem of maximum likelihood estimation (MLE) of the parameters. As mentioned earlier, Zhu [2] extended this model to accommodate zero-inflation. In the following literature review, we will focus on additional work done on INGARCH models without zero-inflation first before moving onto the zero-inflated models.

Weiß [8] extended the previous results on the class of INGARCH models and derived a set of Yule-Walker type equations for the autocorrelation function for the general INGARCH case. Several important theoretical contributions for both the linear and non-linear Poisson

autoregressive models were made by Fokianos et al. [9]. For the model where the conditional mean is a linear function of the past conditional mean and past count, they proved that the maximum likelihood estimates are asymptotically normal. Note that the model they considered is INGARCH (1, 1), but the authors refrain from calling it as such and labelled it as Poisson autoregression because of their stated belief that the GARCH moniker should be reserved for formulations that model variance. In the non-linear case where the conditional mean is a non-linear function of its past values and past counts, the authors established that the intensities of the Poisson distribution at each time point form a geometrically ergodic Markov chain under some general assumptions. Fokianos and Tjøstheim [10] further extended these results and showed that in the log-linear Poisson autoregression case, the maximum likelihood estimates are asymptotically normal, and that covariance matrix of the parameter estimates are consistent. Chen and Lee [11] extended the log-linear Poisson INGARCH formulation by introducing linear effects exogenous variables to the equation for the logarithm of the conditional mean and proposed the log-linear Poisson INGARCHX and negative binomial INGARCHX models. Here X denotes a model with exogenous inputs. The authors suggest that the forecasting performance of the model can be improved by the addition of covariates to the model, and that it helps to properly analyse the causal relationships between exogenous factors and the count time series. The softplus INGARCH model, a non-linear version of INGARCH, was proposed by Weiß, Zhu, and Hoshiyar [12]. This model utilizes the softplus link function as an alternative to the log-link function and thus allowing for negative autocorrelations. We extended this model to include zero-inflation and accommodate an exogenous variable as input. Details of this extension are given in a later section.

Negative binomial (NB), Generalized Poisson (GP), and Double Poisson (DP) are well known discrete distributions that can also be used as an alternative to the Poisson process Zhu [13]. A negative binomial INGARCH model (NB-INGARCH), which is an alternative to the Poisson INGARCH formulation, was proposed and the stationary conditions and the autocorrelation function of the process were obtained by Zhu [14]. Ye, Garcia, Pourahmadi, and Lord [15] allowed the negative binomial INGRACH model from Zhu [14] to incorporate covariates, so that the relationship between a time series of counts and correlated external factors could be properly modelled. Xu, Xie, Goh, and Fu [16] adapted the INARCH formulation and proposed a new dispersed INGARCH (DINARCH) model to handle the conditional overdispersion and underdispersion. Xu et al. [16] also mention that when the dispersion parameter is not constant this model coincides with that of Zhu [14] without the moving average terms.

The idea of modelling the zero-inflated probability in a count data process as a function of covariates can be dated back to Lambert [17]. Zhu [2] while introducing the zero-inflated Poisson, also presented a zero-inflated negative binomial integer-valued GARCH model, but with the assumption of constant zero-inflation probability. In this paper two types of negative binomial distributions are discussed, namely Negative Binomial 1 (NB1), and Negative Binomial 2 (NB2). Chen and Lee [18] investigated the zero-inflated generalized Poisson autoregressive (ZIGP-AR) model of Lee, Lee, and Chen [19] and proposed the zero-inflated generalized Poisson INGARCH (ZIGP-INGARCH) which allows for structural breaks. In the ZIGP-INGARCH model zero-inflation is introduced to the generalized Poisson INGARCH (GP-INGARCH) model of Zhu [20]. Gonçalves, Lopes, and Silva [21] introduced a new class of zero-inflated INGARCH models that included general compound Poisson deviates and named it as the zero-inflated compound Poisson INGARCH (ZICP-INGARCH) process. This model can include both zero-inflated Poisson and zero-inflated negative binomial INGARCH models of Zhu [2].

More recent developments in the INGARCH literature are as follows. Xiong and Zhu [22] and Li, Chen, and Zhu [23] considered the robust estimation methods for INGARCH models.

Liu, Zhu, and Zhu [24] generalized the range of observations from infinite to categorical. Cui, Li and Zhu [25] and Xu and Zhu [26] generalized the INGARCH models from non-negative integer-valued to the integer-valued cases. Lee, Kim and Seok [27] introduced one parameter exponential family INGARCH model with zero-inflation and it was named as ZIEF-IN-GARCH. This new model can accommodate above mention zero-inflated versions of INGARCH models. In addition to that they developed residuals based CUSUM tests which can be used to examine the change points for the proposed model. A summary of various count time series published recently can be found in Davis et al. [28].

None of the above models allow for a time-varying zero-inflation probability. It is imperative that such accommodation should be made because empirical count data time with large number of zero counts can display strong cyclical behavior or seasonality with respect to the observed zero values. Ignoring this time-varying property of the zero-inflation parameter decreases the predictive performance of the model. Recognizing this, Yang [29], discussed the importance of modelling zero-inflation probability as a time-varying function. In the above article, it was assumed that both the zero-inflation and the intensity parameter are driven by the linear combination of past observations of exogenous variables and connects them to the conditional mean of the count via a log-link function. A recently introduced approach to modelling time-varying zero-inflation and the intensity parameter is the adaptive log-linear zero-inflated generalized Poisson model proposed by Xu et al. [30]. The authors assumed that the counts are from a zero-inflated generalized Poisson distribution, with the logarithm of the intensity propagated through a GARCH type model augmented with additional terms of exogenous variables and associated coefficients. All model parameters are assumed to be time-dependent. If this time dependence and the exogenous variable are removed, then the model becomes the log-linear Poisson autoregressive formulation proposed by Fokianos and Tjøstheim [10]. If the time dependence and zero-inflation is removed, then it becomes the log-linear INGARCHX model of Chen and Lee [11]. In applying to monthly crime data from New South Wales, Australia, the authors assumed constant coefficients over a local interval at any given time point $t$ (with the interval adaptively selected from among a prechosen set of nested intervals) and estimated the parameters separately using data from each such interval. In their approach, the shifting of the intervals allows for parameters to change from one interval to another and thus allowing for the zero-inflation as well as other model parameters to vary with time. Note that a version of this model with a time varying exogenous input but non-time varying zero-inflation was included in the comparison of our proposed model against several constant zero-inflation models in an empirical setting.

We propose a different approach based on a generalization of the model proposed by Zhu [2]. In our formulation, it is the zero-inflation probability, rather than the intensity of the Poisson process, that is allowed to be governed by exogenous variables. We also assume that the INGARCH model parameters remain constant over time, allowing for a more parsimonious model that is relatively easier to estimate. While the model proposed by Xu et al. [30] is more flexible, the approach we propose is presented as a simpler and practical alternative, albeit a less sophisticated one. As mentioned earlier, the proposed model can also accommodate the case where the zero-inflation probability is driven by a deterministic function of time or driven by an exogenous variable. In addition, the intensity parameter of the Poisson process is assumed to vary dynamically through a GARCH type model. Thus, the INGARCH part of the proposed model can be viewed as observation driven, in the sense that recursive substitutions can be employed to show that the current intensity of the process conditional on the past is a linear function of past observations and past intensities. Also note that we model the conditional mean rather than the logarithm of the conditional mean as is done in Xu et al. [30].

The remainder of this paper is organized as follows: the first, the Time-Varying Zero-Inflated Poisson INGARCH model (TVZIP-INGARCH) is introduced. Two cases, namely a deterministic cyclically varying zero-inflation component and a model with the zero-inflation parameter driven by an exogenous set of stochastic variables are discussed. A section on parameter estimation procedures is presented next, which is followed by the results of a simulation study. Following which is a section that introduces extensions to the softplus INGARCH model. The performance of this extended model together with additional zero-inflated INGARCH type models are compared to the proposed model in the next section. This section also presents the results of fitting the above models to two empirical data sets. A discussion and conclusions are presented in the final section of this paper.

## The time-varying zero-inflated INGARCH model

As Zhu [2] stated, the probability mass function (pmf) of a zero-inflated Poisson model with parameter vector $(\lambda, \omega)$, with $X$ representing the count, can be written in the following form:

$$P(X = k) = \omega \delta_{k,0} + (1 - \omega)\frac{\lambda^k e^{-\lambda}}{k!}, \ k = 0, 1, 2, \ldots, \text{ where } 0 < \omega < 1 \text{ and}$$

$$\delta_{k,0} = \begin{cases} 1; k = 0 \\ 0; k \neq 0. \end{cases}$$

Further, Zhu [2] presented the mean and the variance of the distribution as follows:

$$E(X) = \lambda(1 - \omega) \text{ and } Var(X) = \lambda(1 - \omega)(1 + \lambda\omega) > E(X) \text{ for } 0 < \omega < 1.$$

Moving on to define the proposed time-varying zero-inflated INGARCH model, assume that $\{X_t : t \in \mathbb{Z}\}$ is a discrete time series of count data, and $\mathbb{F}_{t-1}$ is the sigma field generated by $\{X_l, \lambda_l : l \leq t - 1\}$. The conditional distribution of $X_t$ given $\mathbb{F}_{t-1}$ and $\omega_t$, is assumed to be a zero-inflated Poisson (*ZIP*) with parameter vector $(\mu_t, \omega_t)$ where $\mu_t = (1 - \omega_t)\lambda_t$. Then, $X_t | \mathbb{F}_{t-1}, \omega_t \sim ZIP(\mu_t, \omega_t)$ where,

$$P(X_t = k | \mathbb{F}_{t-1}, \omega_t) = \omega_t \delta_{k,0} + (1 - \omega_t)\frac{\lambda_t^k e^{-\lambda_t}}{k!}. \quad (2)$$

The dynamic propagation of the conditional mean $(\mu_t)$ of the zero-inflated Poisson process is defined by:

$$\mu_t = (1 - \omega_t)\lambda_t,$$

with the intensity parameter $\lambda_t$ of the Poisson process formulated as:

$$\lambda_t = \alpha_0 + \sum_{i=1}^{p} \alpha_i X_{t-i} + \sum_{j=1}^{q} \beta_j \lambda_{t-j},$$

where $\alpha_0 > 0, \alpha_i \geq 0, \beta_j \geq 0, i = 1, 2, 3, \ldots p, j = 1, 2, 3, \ldots, q, p \geq 1, q \geq 1$, and $t \in \mathbb{Z}$. Furthermore, $\omega_t = g(\mathbf{V}_t, \mathbf{\Gamma}) \in (0, 1) \forall t \in \mathbb{Z}$, is a function of variables, propagating over time, which is used to model the time-varying zero-inflation. Note that elements of the vector $\mathbf{V}_t$ may consist of stochastic exogenous variables that vary with time, or it may be a scaler equal to time $t$. In addition, $\mathbf{\Gamma}$ denotes a vector of parameters. It is assumed that $0 < \omega_t < 1$ for all $t \in \mathbb{Z}$. Note that Fokianos et al. [9] defined their linear Poisson autoregressive model for $t \in \mathbb{N}$

instead of $t \in \mathbb{Z}$, with constant initial conditions. The model in (2) can also be defined in a similar manner. It is this alternative formulation that was used in our simulation study.

The above model is denoted by TVZIP-INGARCH $(p, q)$. If $p > 0$ and $q = 0$, then the model becomes a TVZIP-INARCH model with order $p$, denoted by TVZIP-INARCH $(p)$. The conditional mean and conditional variance of $X_t$ given $\mathbb{F}_{t-1}$ and $\omega_t$ are specified by the following equations:

$$E(X_t|\mathbb{F}_{t-1}, \omega_t) = (1 - \omega_t)\lambda_t \text{ and } Var(X_t|\mathbb{F}_{t-1}, \omega_t) = (1 - \omega_t)\lambda_t(1 + \omega_t\lambda_t). \tag{3}$$

The conditional variance to conditional mean ratio, or the dispersion ratio, of TVZIP distribution is:

$$\frac{Var(X_t|\mathbb{F}_{t-1}, \omega_t)}{E(X_t|\mathbb{F}_{t-1}, \omega_t)} = \frac{(1 - \omega_t)\lambda_t(1 + \omega_t\lambda_t)}{(1 - \omega_t)\lambda_t} = (1 + \lambda_t\omega_t) > 1. \tag{4}$$

Note that results (3) and (4) can be derived using the definition of the conditional mean and variance and applying standard procedures utilized in deriving similar quantities related to GARCH type models.

The result in (4) indicates that TVZIP-INGARCH $(p, q)$ can be used to model integer- valued time series with overdispersion if the values of $\omega_t$ and $\lambda_t$ are uniformly bounded below by positive constants.

**Case 1: Zero-inflation driven by a deterministic function of time.** In Case 1, it is assumed that the zero-inflation function $\omega_t = g(\mathbf{V}_t, \mathbf{\Gamma})$ is such $\mathbf{V}_t$ is a scaler equal to $t$. For example, we may assume the function $g$ to be defined as follows:

$$\omega_t = g(\mathbf{V}_t, \mathbf{\Gamma}) = A\sin\left(\frac{2\pi}{s}t\right) + B\cos\left(\frac{2\pi}{s}t\right) + C \tag{5}$$

where $s$ is the seasonal length, and $\mathbf{\Gamma} = \{(A, B, C)^T | A, B, C \in \mathbb{R}\}$.

As mentioned above, the time-varying zero-inflation function $\omega_t = g(\mathbf{V}_t, \mathbf{\Gamma})$ should always be bounded between zero and one. The range of values for $A, B$, and $C$ in (5) that are needed to satisfy the above criterion are derived in S1 Appendix. Note that herein a simple example is used, where the function $g$ consists of a sine function and a cosine function of equal period, but $g$ could be any other function of time that, with proper selection of parameters, can be bounded between zero and one.

**Case 2: Zero-inflated function driven by an exogenous variable.** The proposed model also accommodates the case where the zero-inflation probability is determined by one or more exogenous variables. In this case $g(\mathbf{V}_t, \mathbf{\Gamma})$ is considered a function, with the interval $(0, 1)$ as its range, of the vector of exogenous variables $\mathbf{V}_t$ One example of $g$ is the logistic function. Note that $\mathbf{V}_t$ can be a scaler seasonal autoregressive time series, a vector seasonal time series, or a scaler or vector time series that varies non-seasonally. For illustrative purposes, consider the case where $\mathbf{V}_t$ is a scaler purely seasonal autoregressive time series, denoted by $V_t$, with period $s$, $g$ the logistic function, and $\varepsilon_t$ is a white noise error term. Then we can write,

$$V_t = \eta V_{t-s} + \varepsilon_t, \text{ where } \varepsilon_t \sim WN(0, 1);$$

$$\omega_t = g(V_t, \mathbf{\Gamma}) = \frac{1}{1 + e^{-(\delta_0 + \delta_1 V_t)}}, \tag{6}$$

with $\mathbf{\Gamma} = \{(\delta_0, \delta_1)^T | \delta_0, \delta_1 \in \mathbb{R}\}$.

## Estimation procedure

The use of both the Expectation Maximization (EM) algorithm and Maximum Likelihood (ML) method to estimate the model parameters were developed for the general TVZIP-IN-GARCH ($p$, $q$) case. The TVZIP-INGARCH ($p$, $q$) process is discussed below, and the procedure for the TVZIP-INARCH ($p$) case follows in a similar manner.

**Expectation maximization estimation for the TVZIP-INGARCH ($p$, $q$) process.** Let $X_1$, $X_2$,...,$X_N$ be generated according to the model (2). There are two types of zeros generated by this model. They are the zeroes arising from the Poisson distribution with intensity parameter $\lambda_t$ and the zeroes generated by a Bernoulli process with the probability of obtaining a zero specified by the zero-inflation parameter. Therefore, a given observation can be hypothetically categorized as arising out of a Bernoulli process or as an observation from the Poisson distribution. Let us define $\{Z_t : t \in \mathbb{Z}\}$ to be a Bernoulli random variable such that $Z_t = 1$ if $X_t$ is a generated from the Bernoulli process and $Z_t = 0$ if it is generated by the Poisson distribution. Then,

$$Z_t | \omega_t \sim \text{Bernoulli}(\omega_t) \text{ with } P(Z_t = 1 | \omega_t) = \omega_t \text{ and } P(Z_t = 0 | \omega_t) = (1 - \omega_t).$$

Also, let $\mathbf{Z} = (Z_1, Z_2,...,Z_N)$, $\boldsymbol{\Theta} = (\alpha_0, \alpha_1, \ldots, \alpha_p, \beta_1, \beta_2, \ldots, \beta_q)^T = (\theta_0, \theta_1, \ldots, \theta_{p+q})^T$ and $\omega_t = g(\mathbf{V}_t, \boldsymbol{\Gamma})$. Note that $\boldsymbol{\Gamma} = (\gamma_0, \gamma_1, \ldots, \gamma_r)^T$, where $r$ is the dimension of the vector $\mathbf{V}_t$. For notational simplicity, we define the composite parameter vector $\boldsymbol{\Phi} = (\boldsymbol{\Gamma}^T, \boldsymbol{\Theta}^T)^T = (\phi_1, \phi_2, \ldots \ldots, \phi_{r+p+q+2})^T \subseteq \mathbb{R}^{r+p+q+2}$, with the original parameters renamed as $\phi_k$, $k = 1, 2,...,r+p+q+2$. This simplified notation is used in situations where generic statements are made without reference to a specific portion of (2).

Paralleling the derivations in Zhu [2], the conditional log likelihood can be written as (see S2 Appendix for details),

$$l(\boldsymbol{\Phi}) = \sum_{t=p+1}^{N} \{Z_t \log(\omega_t) + (1 - Z_t)[\log(1 - \omega_t) + X_t \log(\lambda_t) - \lambda_t - \log(X_t!)]\}. \tag{7}$$

The first derivatives of the conditional log likelihood function (7) with respect to $\boldsymbol{\Gamma} = (\gamma_0, \gamma_1, \ldots, \gamma_r)^T$ and $\boldsymbol{\Theta} = (\theta_0, \theta_1, \ldots, \theta_{p+q})^T$ are as follows:

$$\frac{dl(\boldsymbol{\Phi})}{d\gamma_i} = \frac{dl(\boldsymbol{\Phi})}{d\omega_t} \frac{d\omega_t}{d\gamma_i} = \sum_{t=p+1}^{N} \left\{ \frac{Z_t}{\omega_t} - \frac{(1 - Z_t)}{(1 - \omega_t)} \right\} \frac{d\omega_t}{d\gamma_i}, i = 0, 1, \ldots, r, \tag{8}$$

$$\frac{dl(\boldsymbol{\Phi})}{d\theta_j} = \frac{dl(\boldsymbol{\Phi})}{d\lambda_t} \frac{d\lambda_t}{d\theta_j} = \sum_{t=p+1}^{N} (1 - Z_t) \left\{ \frac{X_t}{\lambda_t} - 1 \right\} \frac{d\lambda_t}{d\theta_j}, j = 0, 1, \ldots, p + q. \tag{9}$$

Finally, by combining (8) and (9) the first derivative of the conditional log likelihood function with respect to $\Phi$ is given by:

$$\frac{dl(\boldsymbol{\Phi})}{d\phi_k} = \begin{cases} \dfrac{dl(\boldsymbol{\Phi})}{d\omega_t} \dfrac{d\omega_t}{d\phi_k}; \phi_k \in \boldsymbol{\Gamma} \\ \dfrac{dl(\boldsymbol{\Phi})}{d\lambda_t} \dfrac{d\lambda_t}{d\phi_k}; \phi_k \in \boldsymbol{\Theta} \end{cases}. \tag{10}$$

The two-step (E step and M step) Expectation Maximization algorithm is now used to estimate the parameter vector $\boldsymbol{\Phi} = (\boldsymbol{\Gamma}^T, \boldsymbol{\Theta}^T)^T$. Let $\tau_t = E(Z_t | X_t, \boldsymbol{\Phi})$ and we replace $Z_t$ by $\hat{Z}_t = \tau_t$

and define $\mathbf{Z} = (Z_1, Z_2, \ldots, Z_N)^T$. Following this replacement of $\mathbf{Z}$ in the conditional log likelihood function, $l(\mathbf{\Phi}, \hat{\mathbf{Z}})$ is maximized.

**E Step:** Determine $\tau_t$ using the equation

$$\tau_t = \begin{cases} \dfrac{\omega_t}{\omega_t + (1 - \omega_t)e^{-\lambda_t}} & : X_t = 0, \\ 0 & : X_t > 0. \end{cases}$$

**M Step:** After $Z_t$ is replaced by its estimate, we proceed to maximize $l(\mathbf{\Phi}, \hat{Z})$. First set $\frac{dl(\mathbf{\Phi})}{d\phi_k} = 0$, for $k = 1, 2, \ldots, r+p+q+2$.

If, $\hat{\mathbf{\Phi}}$, the solution to the system of equations in (10) exists, then $\mathbf{S}(\hat{\mathbf{\Phi}}) = \mathbf{0}$, where $\mathbf{S}(\mathbf{\Phi})$ is the Fisher's score matrix, and $\hat{\mathbf{\Phi}}$ is the vector that maximizes the log likelihood, thus providing us with the estimate of the parameter vector $\mathbf{\Phi} = (\mathbf{\Gamma^T}, \mathbf{\Theta^T})^T$.

Since a closed form solution does not exist, we require an iterative procedure to find the estimates. Let us consider the first order Taylor expansion of $\mathbf{S}(\tilde{\mathbf{\Phi}})$ evaluated at the value $\tilde{\mathbf{\Phi}}$ around the initial parameter values $\mathbf{\Phi_0}$, yielding $\mathbf{S}(\tilde{\mathbf{\Phi}}) \approx \mathbf{S}(\mathbf{\Phi_0}) + (\tilde{\mathbf{\Phi}} - \mathbf{\Phi_0})\frac{d\mathbf{S}(\mathbf{\Phi})}{d\mathbf{\Phi}}|_{\mathbf{\Phi}=\mathbf{\Phi_0}}$. We also let the matrix of the second derivatives of the log likelihood function to be defined as $\mathbf{H}(\mathbf{\Phi}) = \frac{d^2 l(\mathbf{\Phi})}{d\mathbf{\Phi}d\mathbf{\Phi}^T} = \frac{d\mathbf{S}(\mathbf{\Phi})}{d\mathbf{\Phi}}$.

From the above, we obtain the first order approximation $\tilde{\mathbf{\Phi}} = \mathbf{\Phi_0} - \mathbf{H}^{-1}(\mathbf{\Phi_0})\mathbf{S}(\mathbf{\Phi_0})$, and this result provides the standard Newton-Raphson algorithm. For an appropriately chosen initial value $\hat{\mathbf{\Phi}}^{(0)}$, the above Newton-Raphson algorithm can be used to obtain a sequence of improved estimates recursively. The improved estimates at $i^{\text{th}}$ iteration are updated as the initial values for the next iteration as follows:

$$\hat{\mathbf{\Phi}}^{(i+1)} = \hat{\mathbf{\Phi}}^{(i)} - \mathbf{H}^{-1}(\hat{\mathbf{\Phi}}^{(i)})\mathbf{S}(\hat{\mathbf{\Phi}}^{(i)}).$$

This process is repeated until the differences between successive estimates are sufficiently close to zero. In our study, convergence of the EM procedure was determined by using the criterion:

$$\left| \frac{(\hat{\phi}_j^{(i+1)} - \hat{\phi}_j^{(i)})}{\hat{\phi}_j^{(i)}} \right| \leq 10^{-6}.$$

**Maximum likelihood estimation for the TVZIP-INGARCH ($p, q$) process.** The conditional likelihood function $L(\Phi)$ of the TVZIP-INGARCH model (2) is,

$$L(\mathbf{\Phi}) = \prod_{X_t=0} [\omega_t + (1 - \omega_t)e^{-\lambda_t}] \times \prod_{X_t>0} \left[ (1 - \omega_t)\frac{\lambda_t^{X_t} e^{-\lambda_t}}{X_t!} \right]. \quad (11)$$

The conditional log likelihood function, $l(\Phi)$ obtained from (11) is given by

$$l(\mathbf{\Phi}) = \sum_{X_t=0} \log(\omega_t + (1 - \omega_t)e^{-\lambda_t}) + \sum_{X_t>0} [\log(1 - \omega_t) + X_t \log(\lambda_t) - \lambda_t - \log(X_t!)]. \quad (12)$$

Let $P_{0,t} = \omega_t + (1 - \omega_t)e^{-\lambda_t}$ and $I(X_t = 0) = x_{0,t}$. Then,

$$\frac{dl(\mathbf{\Phi})}{d\omega_t} = \sum_{t=p+1}^{N} \left[ \frac{x_{0,t}(1 - e^{-\lambda_t})}{P_{0,t}} - \frac{1 - x_{0,t}}{1 - \omega_t} \right], \quad (13)$$

and

$$\frac{dl(\mathbf{\Phi})}{d\lambda_t} = \sum_{t=p+1}^{N} \left[ \frac{x_{0,t}(\omega_t - 1)e^{-\lambda_t}}{P_{0,t}} + \frac{(1 - x_{0,t})(X_t - \lambda_t)}{\lambda_t} \right]. \tag{14}$$

The first derivatives of the conditional log likelihood function (12) are as follows,

$$\frac{dl(\mathbf{\Phi})}{d\gamma_i} = \frac{dl(\mathbf{\Phi})}{d\omega_t}\frac{d\omega_t}{d\gamma_i} = \sum_{t=p+1}^{N} \left[ \frac{x_{0,t}(1 - e^{-\lambda_t})}{P_{0,t}} - \frac{1 - x_{0,t}}{1 - \omega_t} \right] \frac{d\omega_t}{d\gamma_i}, i = 0, 1, \ldots, r, \tag{15}$$

$$\frac{dl(\mathbf{\Phi})}{d\theta_j} = \frac{dl(\mathbf{\Phi})}{d\lambda_t}\frac{d\lambda_t}{d\theta_j} = \sum_{t=p+1}^{N} \left[ \frac{x_{0,t}(\omega_t - 1)e^{-\lambda_t}}{P_{0,t}} + \frac{(1 - x_{0,t})(X_t - \lambda_t)}{\lambda_t} \right] \frac{d\lambda_t}{d\theta_j}, j$$
$$= 0, 1, \ldots, p + q, \tag{16}$$

$$\frac{dl(\mathbf{\Phi})}{d\phi_k} = \begin{cases} \frac{dl(\mathbf{\Phi})}{d\omega_t}\frac{d\omega_t}{d\phi_k}; \phi_k \in \mathbf{\Gamma} \\ \frac{dl(\mathbf{\Phi})}{d\lambda_t}\frac{d\lambda_t}{d\phi_k}; \phi_k \in \mathbf{\Theta} \end{cases}.$$

We can use Newton-Raphson (NR) iterative procedure to obtain the maximum likelihood estimated for the Eq (12) by setting $\frac{dl(\mathbf{\Phi})}{d\phi_k} = 0$ for all $k$. With a reasonable initial starting value $\hat{\mathbf{\Phi}}^{(0)}$, the $i$th iteration is calculated using $\hat{\mathbf{\Phi}}^{(i+1)} = \hat{\mathbf{\Phi}}^{(i)} - \mathbf{H}^{-1}(\hat{\mathbf{\Phi}}^{(i)})\mathbf{S}(\hat{\mathbf{\Phi}}^{(i)})$, where $\mathbf{S}(\hat{\mathbf{\Phi}}) = \frac{dl(\mathbf{\Phi})}{d\mathbf{\Phi}}|_{\mathbf{\Phi}=\hat{\mathbf{\Phi}}})$ and $\mathbf{H}(\hat{\mathbf{\Phi}}) = \frac{dl(\mathbf{\Phi})}{d\mathbf{\Phi}d\mathbf{\Phi}^{\mathbf{T}}}|_{\mathbf{\Phi}=\hat{\mathbf{\Phi}}}$ for $k = 1,2,\ldots,r+p+q+2$. We stop the algorithm once pre specified convergence criteria is satisfied.

Both the EM and the ML estimation procedures require careful selection of initial values of the parameter vector $\hat{\mathbf{\Phi}}^{(0)} = (\hat{\mathbf{\Theta}}^{(0)}, \hat{\mathbf{\Gamma}}^{(0)})$ to initiate the iterative algorithm. As described in Fokianos et al. [9], the starting value $\hat{\mathbf{\Theta}}^{(0)}$ is computed by fitting a ARMA ($p$, $q$) model to the data. The initial value for the parameter in $\hat{\mathbf{\Gamma}}^{(0)}$ is selected from a range of values on a grid that satisfies the conditions of the time-varying zero inflation function. As discussed in Weiß [31], when $q>0$ the parameter estimates of a INGARCH ($p$, $q$) are extremely sensitive to the choice of the initial conditional mean. In other words, inappropriate initialization of $\hat{\lambda}_1^{(0)}$ may exhibit a significant effect on $\hat{\mathbf{\Theta}}$. To address this issue, two courses of actions for computing $\hat{\lambda}_1^{(0)}$ are suggested. One is to treat $\hat{\lambda}_1^{(0)}$ as a parameter during estimation such that $\hat{\lambda}_1^{(0)} = \hat{\alpha}_0^{(0)}$ and the other is to use a fixed value, for example $\hat{\lambda}_1 = \bar{x}$. The results (see S3 Appendix) show that both initialization procedures produce relatively similar estimates, with a slightly better fit for the data observed for the initialization case with $\hat{\lambda}_1^{(0)} = \hat{\alpha}_0^{(0)}$. Therefore, the initial values of this study are specified according to $\hat{\lambda}_1^{(0)} = \hat{\alpha}_0^{(0)}$ case.

## Simulation study

We investigated the finite sample performance of estimators using a simulation study. The *poissrnd* function of MATLAB software was employed to generate the relevant data, based on recursively computed conditional intensity parameter. In order to initiate the recurve process, the intensity at time $t = 0$ and count data at times $t \leq 0$ were set to zero (i.e., $\lambda_0 = 0$ and $X_l = 0$ for $l \leq 0$). For time periods $t \geq 1$, $X_t$ was generated as follows. For each $t \in \mathbb{N}$ let $U_t$ be a random variable generated from a uniform (0,1) distribution and let $\omega_t$ be the zero-inflated probability

at time $t$. Then, $X_t$ was set to zero if $U_t \leq \omega_t$, Otherwise $X_t$ was generated from the Poisson distribution with intensity parameter $\lambda_t$, where $\lambda_t$ was updated recursively using Eq (2). The process was repeated until the complete time series of length $N$ was generated. Note that this is the same procedure Zhu [2] employed to generate data for his simulation study. Lengths of the time series studied were set to $N = 120$ and $N = 360$, and thousand ($m = 1000$) simulations runs were carried out for each parameter and sample size combination. We carried out two separate sets of simulation studies based on the two types of zero-inflation function introduced in Section 2. The profile log likelihood function given in Eq (12) was maximized using the constrained nonlinear optimization function *fmincon* in MATLAB. The zero-inflation probability ($\omega_t = g(\mathbf{V}_t, \mathbf{\Gamma})$) was allowed to vary cyclically as a deterministic function of time or to be driven by an exogenous variable. Following Zhu [2], the Mean Absolute Deviation Error (MADE) was utilized as the evaluation criterion. The MADE is defined as, $\frac{1}{m}\sum_{j=1}^{m}|\hat{\phi}_j - \phi|$ where $m$ is the number of replications and $\phi \in \mathbf{\Phi} = (\mathbf{\Gamma}^T, \mathbf{\Theta}^T)^T$ is the true value while $\hat{\phi}_j$ is the estimated value of $\phi$ at $j$th replication run. In addition, computational effort is expressed by CPU time (in seconds) and it is used as a criterion to evaluate the performance. Simulation results for Case 1 and Case 2 are reported below.

**Simulation results for Case 1: Deterministic sinusoidal zero-inflation function.** In this portion of the simulation study, the sinusoidal zero-inflated function $\omega_t = g(\mathbf{V}_t, \mathbf{\Gamma})$ expressed in Eq (5) was used to generate cyclically varying zero-inflation probabilities between zero and one. The following constraints are set to the parameters in the vector $\mathbf{\Gamma} = \{(A, B, C)^T | A, B, C \in \mathbb{R}\} : C = \sqrt{A^2 + B^2} + \delta$, where $|A| \leq \frac{1}{2} - \delta$, $|B| \leq \frac{1}{2} - \delta$, and a constant $\delta \in (0, \frac{1}{2})$ such that $\sqrt{A^2 + B^2} \leq \frac{1}{2} - \delta$. Note that the above constraints, with $\delta = 0.0001$, were applied in our simulation study in order to bound the zero-inflation probabilities between 0 and 1. A very small value for $\delta$ was selected to allow wider bounds for $\sqrt{A^2 + B^2}$, $|A|$, and $|B|$.

Tables 1 through 3 provide the simulation results for the MLE estimation technique, while Tables 4 through 6 provide simulation results for the case where estimates were obtained using the EM algorithm. The frequency of the sinusoidal wave was set at $s = 12$, mimicking a 12-month cycle present in monthly data. The parameter vector for the simulation study was expressed as $\mathbf{\Phi} = (A, B, \alpha_0, \alpha_1, \alpha_2, \beta_1)^T$, where $A$ and $B$ are the parameters in the sinusoidal model while $(\alpha_0, \alpha_1)$, $(\alpha_0, \alpha_1, \alpha_2)$, and $(\alpha_0, \alpha_1, \beta_1)$ are the parameter combinations in TVZIP-I-NARCH (1), TVZIP-INARCH (2), and TVZIP-INGARCH (1, 1) models, respectively. The parameter combination of $\mathbf{\Gamma} = (A, B)^T$ was set at $(0.10, 0.10)^T$, $(0.25, -0.20)^T$, and $(-0.35, -0.30)^T$, representing minimal to minimal, minimal to moderate, and minimal to maximum zero-inflation ranges. The following models were considered:

(A) TVZIP-INARCH (1) models: $\mathbf{\Phi} = (A, B, \alpha_0, \alpha_1)^T$

 A1. $(0.10, \ 0.10, \ 1.00, \ 0.40)^T$
 A2. $(-0.25, \ -0.25, \ 2.00, \ 0.50)^T$
 A3. $(-0.35, \ -0.30, \ 1.00, \ 0.70)^T$

(B) TVZIP-INARCH (2) models: $\mathbf{\Phi} = (A, \ B, \ \alpha_0, \ \alpha_1, \ \alpha_2)^T$

 B1. $(0.10, \ 0.10, \ 1.00, \ 0.20, \ 0.20)^T$
 B2. $(-0.25, \ -0.25, \ 2.00, \ 0.30, \ 0.20)^T$
 B3. $(-0.35, \ -0.30, \ 1.00, \ 0.40, \ 0.30)^T$

(C) TVZIP-INGARCH (1,1) models: $\mathbf{\Phi} = (A, \ B, \ \alpha_0, \ \alpha_1, \ \beta_1)^T$

**Table 1.** Means of MLE estimates MADE (within parentheses), and computational efforts (CPU time) for TVZIP-INARCH (1) models where zero-inflation is driven by a sinusoidal function.

| Model | $N$ | $\hat{A}$ | $\hat{B}$ | $\hat{\alpha}_0$ | $\hat{\alpha}_1$ | CPU time (seconds) |
|---|---|---|---|---|---|---|
| | True values | 0.10 | 0.10 | 1.00 | 0.40 | |
| A1 | 120 | 0.0894 (0.0560) | 0.0846 (0.0565) | 1.0473 (0.1434) | 0.3712 (0.0903) | 45.30 |
| | 360 | 0.0986 (0.0321) | 0.0951 (0.0299) | 1.0172 (0.0802) | 0.3917 (0.0479) | 219.98 |
| | True values | -0.25 | -0.25 | 2.00 | 0.50 | |
| A2 | 120 | -0.2487 (0.0401) | -0.2476 (0.0399) | 2.0467 (0.2210) | 0.4800 (0.0873) | 43.16 |
| | 360 | -0.2504 (0.0225) | -0.2478 (0.0223) | 2.0203 (0.1311) | 0.4925 (0.0484) | 202.14 |
| | True values | -0.35 | -0.30 | 1.00 | 0.70 | |
| A3 | 120 | -0.3468 (0.0458) | -0.2958 (0.0482) | 1.0370 (0.1621) | 0.6675 (0.1164) | 46.63 |
| | 360 | -0.3508 (0.0250) | -0.2969 (0.0263) | 1.0178 (0.0963) | 0.6875 (0.0629) | 194.83 |

C1. $(0.10, \ 0.10, \ 1.00, \ 0.20, \ 0.20)^T$

C2. $(-0.25, \ -0.25, \ 2.00, \ 0.30, \ 0.20)^T$

C3. $(-0.35, \ -0.30, \ 1.00, \ 0.40, \ 0.30)^T$

Note that the two estimations procedures were run on identical simulation samples for each model, parameter, and sample size combinations and hence variations due to sampling error will not be seen when comparing across estimation methods. The following tables provide summary data from the simulation runs, with the actual parameter values, estimated values, the mean absolute deviation (MADE) between them, and the CPU time taken by the estimation procedures.

The above simulation results show that the MLE and EM procedures produced almost identical means of estimates for the parameters in TVZIP-INARCH (1) and TVZIP-INARCH (2) models. For example, the means of estimates in Table 1 are almost identical to the corresponding means of estimates in Table 4. The MADE values for corresponding estimates are also almost identical across the two estimation methods. This similarity extends to means of corresponding parameter estimates across Tables 2 and 5 as well. Even in cases where the means of estimates are not identical, they are extremely close. For instance, for Model A1 with $N = 120$ and the true value of $\alpha_0$ equals to 1.00, the mean of the MLE estimates for this parameter is 1.0473 while the mean of the EM estimates is 1.0472.

However, for the TVZIP-INGARCH (1, 1) process, there are relatively larger differences between means of estimates for the MLE and EM methods. For example, when $N = 120$ with true parameter values of $\alpha_0 = 1.00$, $\alpha_1 = 0.20$, $\beta_1 = 0.20$, the means of parameters estimates are

**Table 2.** Means of MLE estimates and MADE (within parentheses), and computational efforts (CPU time) for TVZIP-INARCH (2) models where zero-inflation is driven by a sinusoidal function.

| Model | $N$ | $\hat{A}$ | $\hat{B}$ | $\hat{\alpha}_0$ | $\hat{\alpha}_1$ | $\hat{\alpha}_2$ | CPU time (seconds) |
|---|---|---|---|---|---|---|---|
| | True values | 0.10 | 0.10 | 1.00 | 0.20 | 0.20 | |
| B1 | 120 | 0.0874 (0.0557) | 0.0840 (0.0537) | 1.0528 (0.1601) | 0.1877 (0.0873) | 0.1765 (0.0863) | 87.41 |
| | 360 | 0.0955 (0.0317) | 0.0971 (0.0290) | 1.0239 (0.0945) | 0.1943 (0.0496) | 0.1906 (0.0503) | 447.09 |
| | True values | -0.25 | -0.25 | 2.00 | 0.30 | 0.20 | |
| B2 | 120 | -0.2485 (0.0430) | -0.2476 (0.0395) | 2.0524 (0.2563) | 0.2842 (0.0952) | 0.1906 (0.0931) | 77.31 |
| | 360 | -0.2514 (0.0234) | -0.2470 (0.0224) | 2.0254 (0.1478) | 0.2989 (0.0500) | 0.1901 (0.0536) | 337.33 |
| | True values | -0.35 | -0.30 | 1.00 | 0.40 | 0.30 | |
| B3 | 120 | -0.3478 (0.0455) | -0.2940 (0.0486) | 1.0357 (0.1712) | 0.3840 (0.1305) | 0.2781 (0.1325) | 69.95 |
| | 360 | -0.3510 (0.0282) | -0.2974 (0.0273) | 1.0149 (0.1004) | 0.3975 (0.0709) | 0.2914 (0.0762) | 357.02 |

**Table 3.** Means of MLE estimates, MADE (within parentheses), and computational efforts (CPU time) for TVZIP-INGARCH (1, 1) models where zero-inflation is driven by a sinusoidal function.

| Model | N | $\hat{A}$ | $\hat{B}$ | $\hat{\alpha}_0$ | $\hat{\alpha}_1$ | $\hat{\beta}_1$ | CPU time (seconds) |
|---|---|---|---|---|---|---|---|
| | True values | 0.10 | 0.10 | 1.00 | 0.20 | 0.20 | |
| C1 | 120 | 0.0938 (0.0522) | 0.0882 (0.0543) | 0.9305 (0.2306) | 0.2224 (0.0788) | 0.2105 (0.1407) | 998.63 |
| | 360 | 0.1057 (0.0300) | 0.0983 (0.0295) | 0.9464 (0.1605) | 0.2296 (0.0553) | 0.1905 (0.1107) | 3857.80 |
| | True values | -0.25 | -0.25 | 2.00 | 0.30 | 0.20 | |
| C2 | 120 | -0.2523 (0.0396) | -0.2470 (0.0401) | 1.7525 (0.3688) | 0.3614 (0.0976) | 0.1882 (0.1132) | 555.03 |
| | 360 | -0.2516 (0.0231) | -0.2500 (0.0221) | 1.8064 (0.2357) | 0.3843 (0.0884) | 0.1539 (0.0859) | 2034.50 |
| | True values | -0.35 | -0.30 | 1.00 | 0.40 | 0.30 | |
| C3 | 120 | -0.3591 (0.0415) | -0.2934 (0.0454) | 0.9287 (0.2298) | 0.4668 (0.1291) | 0.2249 (0.1631) | 792.75 |
| | 360 | -0.3592 (0.0259) | -0.2983 (0.0265) | 0.9321 (0.1532) | 0.4961 (0.1085) | 0.2008 (0.1357) | 3207.00 |

$\hat{\alpha}_0 = 0.9305$, $\hat{\alpha}_1 = 0.2224$, $\hat{\beta}_1 = 0.2105$ for the MLE method (Table 3, row 1) while EM mean estimates are $\hat{\alpha}_0 = 0.9334$, $\hat{\alpha}_1 = 0.2215$, $\hat{\beta}_1 = 0.2153$ (Table 6, row 1). Generally, the larger sample size produced means of estimates closer to their true value. An exception to this is observed in the case of TVZIP-INGARCH (1, 1), where the means of estimates for $(\alpha_1,\beta_1)$ did not improve with increasing sample size. For instance, when $N = 120$ with true values of $\alpha_1 = 0.40$, $\beta_1 = 0.30$, the means of the corresponding MLE estimates are 0.4668 and 0.2249 respectively (see Table 3, Model C3), but when the sample size increases to 360, the means of the corresponding estimates changed to 0.4961 and 0.2008 respectively, which is a movement in the wrong direction. The MADE values decreased consistently with increasing sample size for both the MLE and EM estimation methods. Another relevant observation is that the simulation results for the INGARCH portion $(\alpha_1,\beta_1)$ of the model behave similar to the results obtained by Zhu [2] in the sense that the mean of the estimates are not very close to the true values even with higher sample sizes. We investigated whether this relative inaccuracy of the estimates is due to the initial values that were used for these two parameters. This was done by creating a grid of potential initial values centered on the original initial values obtained by fitting an ARAMA to the data. This approach did not improve the results and thus we reverted to using the original initial values. It is possible that many combinations of $\hat{\alpha}$ and $\hat{\beta}$ values provide very similar likelihoods and thus the algorithm does not converge to the true values efficiently. Another important observation is that the EM algorithm took more CPU time than the MLE method across all parameter and model combinations.

**Table 4.** Means of EM estimates, MADE (within parentheses), and computational efforts (CPU time) for TVZIP-INARCH (1) models where zero-inflation is driven by a sinusoidal function.

| Model | N | $\hat{A}$ | $\hat{B}$ | $\hat{\alpha}_0$ | $\hat{\alpha}_1$ | CPU time (seconds) |
|---|---|---|---|---|---|---|
| | True values | 0.10 | 0.10 | 1.00 | 0.40 | |
| A1 | 120 | 0.0898 (0.0557) | 0.0850 (0.0561) | 1.0472 (0.1433) | 0.3712 (0.0903) | 1380.10 |
| | 360 | 0.0986 (0.0321) | 0.0951 (0.0299) | 1.0172 (0.0802) | 0.3917 (0.0479) | 7440.80 |
| | True values | -0.25 | -0.25 | 2.00 | 0.50 | |
| A2 | 120 | -0.2487 (0.0401) | -0.2476 (0.0398) | 2.0467 (0.2210) | 0.4800 (0.0873) | 326.72 |
| | 360 | -0.2504 (0.0225) | -0.2478 (0.0222) | 2.0203 (0.1311) | 0.4925 (0.0484) | 1974.30 |
| | True values | -0.35 | -0.30 | 1.00 | 0.70 | |
| A3 | 120 | -0.3468 (0.0458) | -0.2958 (0.0482) | 1.0370 (0.1621) | 0.6675 (0.1164) | 507.64 |
| | 360 | -0.3508 (0.0250) | -0.2969 (0.0263) | 1.0178 (0.0963) | 0.6875 (0.0629) | 2738.31 |

**Table 5.** Means of EM estimates, MADE (within parentheses), and computational efforts (CPU time) for TVZIP-INARCH (2) models where zero-inflation is driven by a sinusoidal function.

| Model | N | $\hat{A}$ | $\hat{B}$ | $\hat{\alpha}_0$ | $\hat{\alpha}_1$ | $\hat{\alpha}_2$ | CPU time (seconds) |
|---|---|---|---|---|---|---|---|
| | True values | **0.10** | **0.10** | **1.00** | **0.20** | **0.20** | |
| B1 | 120 | 0.0875 (0.0556) | 0.0841 (0.0536) | 1.0528 (0.1601) | 0.1877 (0.0873) | 0.1765 (0.0863) | 1736.20 |
| | 360 | 0.0955 (0.0317) | 0.0972 (0.0290) | 1.0239 (0.0945) | 0.1943 (0.0496) | 0.1906 (0.0503) | 8215.50 |
| | True values | **-0.25** | **-0.25** | **2.00** | **0.30** | **0.20** | |
| B2 | 120 | -0.2485 (0.0430) | -0.2476 (0.0395) | 2.0524 (0.2563) | 0.2842 (0.0952) | 0.1906 (0.0931) | 456.06 |
| | 360 | -0.2514 (0.0234) | -0.2470 (0.0224) | 2.0254 (0.1478) | 0.2989 (0.0500) | 0.1901 (0.0536) | 2107.10 |
| | True values | **-0.35** | **-0.30** | **1.00** | **0.40** | **0.30** | |
| B3 | 120 | -0.3480 (0.0453) | -0.2940 (0.0485) | 1.0357 (0.1712) | 0.3840 (0.1305) | 0.2781 (0.1324) | 700.98 |
| | 360 | -0.3510 (0.0282) | -0.2974 (0.0273) | 1.0149 (0.1004) | 0.3975 (0.0709) | 0.2914 (0.0762) | 3341.70 |

## Simulation study for Case 2: Zero-inflation function driven by an exogenous variable

In this part of the study, we allow the exogenous variable to generate zeros through a logistic model as described in Eq (6). The parameter vector for the simulation study under this scenario is $\mathbf{\Phi} = (\delta_0, \delta_1, \alpha_0, \alpha_1, \alpha_2, \beta_1)^T$, where $\delta_0$ and $\delta_1$ are the parameters in the logistic part of the model, while $(\alpha_0, \alpha_1)$, $(\alpha_0, \alpha_1, \alpha_2)$, and $(\alpha_0, \alpha_1, \beta_1)$ are the parameter combination for TVZIP-INARCH (1), TVZIP-INARCH (2), and TVZIP-INGARCH (1, 1) models, respectively. The parameter combinations of $\delta_0$ and $\delta_1$ were set to (-2, 0), (-1, -1) and (2, 1), representing three types of changes in zero-inflation probability with respect to the exogenous variable. These are no change ($\delta_1 = 0$), decrease ($\delta_1 < 0$), and increase ($\delta_1 > 0$)in the zero-inflation probability with increasing values of the exogenous variable. We generated an exogenous stationary AR (12) time series given in Eq (6) using $\eta = 0.25$. The following models were considered:

(A) TVZIP-INARCH (1) models: $\mathbf{\Phi} = (\delta_0, \delta_1, \alpha_0, \alpha_1)^T$

 A1. $(-2.00, \ 0.00, \ 1.00, \ 0.40)^T$

 A2. $(-1.00, \ -1.00, \ 2.00, \ 0.50)^T$

 A3. $(2.00, \ 1.00, \ 1.00, \ 0.70)^T$

(B) TVZIP-INARCH (2) models: $\mathbf{\Phi} = (\delta_0, \ \delta_1, \ \alpha_0, \ \alpha_1, \ \alpha_2)^T$

 B1. $(-2.00, \ 0.00, \ 1.00, \ 0.20, \ 0.20)^T$

**Table 6.** Means of EM estimates, MADE (within parentheses), and computational efforts (CPU time) for TVZIP-INGARCH (1, 1) models where zero-inflation is driven by a sinusoidal function.

| Model | N | $\hat{A}$ | $\hat{B}$ | $\hat{\alpha}_0$ | $\hat{\alpha}_1$ | $\hat{\beta}_1$ | CPU time (seconds) |
|---|---|---|---|---|---|---|---|
| | True values | **0.10** | **0.10** | **1.00** | **0.20** | **0.20** | |
| C1 | 120 | 0.0957 (0.0538) | 0.0901 (0.0561) | 0.9334 (0.2229) | 0.2215 (0.0793) | 0.2153 (0.1418) | 1649.80 |
| | 360 | 0.1070 (0.0314) | 0.0990 (0.0301) | 0.9433 (0.1603) | 0.2292 (0.0556) | 0.1952 (0.1122) | 8232.90 |
| | True Values | **-0.25** | **-0.25** | **2.00** | **0.30** | **0.20** | |
| C2 | 120 | -0.2519 (0.0397) | -0.2468 (0.0405) | 1.7496 (0.3704) | 0.3614 (0.0974) | 0.1923 (0.1162) | 581.46 |
| | 360 | -0.2516 (0.0231) | -0.2500 (0.0221) | 1.8070 (0.2347) | 0.3843 (0.0884) | 0.1539 (0.0859) | 2157.70 |
| | True Values | **-0.35** | **-0.30** | **1.00** | **0.40** | **0.30** | |
| C3 | 120 | -0.3585 (0.0414) | -0.2930 (0.0453) | 0.9238 (0.2217) | 0.4672 (0.1286) | 0.2356 (0.1674) | 880.33 |
| | 360 | -0.3591 (0.0258) | -0.2984 (0.0264) | 0.9349 (0.1495) | 0.4961 (0.1086) | 0.2017 (0.1367) | 3617.00 |

**Table 7.** Means of MLE Estimates, MADE (within parentheses), and computational efforts (CPU time) for TVZIP-INARCH (1) models where zero-inflation is driven by an exogenous variable.

| Model | $N$ | $\hat{\delta}_0$ | $\hat{\delta}_1$ | $\hat{\alpha}_0$ | $\hat{\alpha}_1$ | CPU time (seconds) |
|---|---|---|---|---|---|---|
| | True values | -2.00 | 0.00 | 1.00 | 0.40 | |
| A1 | 120 | -2.4301 (0.7134) | -0.0331 (0.5515) | 1.0169 (0.1381) | 0.3808 (0.0837) | 39.19 |
| | 360 | -2.0982 (0.3258) | -0.0162 (0.2559) | 1.0143 (0.0817) | 0.3892 (0.0487) | 138.56 |
| | True values | -1.00 | -1.00 | 2.00 | 0.50 | |
| A2 | 120 | -1.0469 (0.2384) | -1.0714 (0.2583) | 2.0193 (0.2143) | 0.4864 (0.0836) | 37.28 |
| | 360 | -1.0209 (0.1338) | -1.0182 (0.1400) | 2.0073 (0.1199) | 0.4932 (0.0476) | 145.00 |
| | True values | 2.00 | 1.00 | 1.00 | 0.70 | |
| A3 | 120 | 2.0032 (0.4366) | 1.1757 (0.4312) | 0.9901 (0.2988) | 0.4603 (0.3699) | 38.56 |
| | 360 | 1.9976 (0.2197) | 1.0365 (0.1947) | 1.0037 (0.1694) | 0.5662 (0.2739) | 134.84 |

$$B2. \ (-1.00, \ -1.00, \ 2.00, \ 0.30, \ 0.20)^T$$
$$B3. \ (2.00, \ 1.00, \ 1.00, \ 0.40, \ 0.30)^T$$

(C) TVZIP-INGARCH (1,1) models: $\mathbf{\Phi} = (\delta_0, \ \delta_1, \ \alpha_0, \ \alpha_1, \ \beta_1)^T$

$$C1. \ (-2.00, \ 0.00, \ 1.00, \ 0.20, \ 0.20)^T$$
$$C2. \ (-1.00, \ -1.00, \ 2.00, \ 0.30, \ 0.20)^T$$
$$C3. \ (2.00, \ 1.00, \ 1.00, \ 0.40, \ 0.30)^T$$ Tables 7 through 9 provide the simulation results for the MLE estimation techniques, while Tables 10 through 12 provide EM algorithm related simulation results.

In general, the EM algorithm takes more computational effort to estimate parameters when compared to the same model parameter estimation done by the MLE procedure. Both the MLE and EM methods produced fairly accurate estimates for the parameters across both TVZIP-INARCH ($p$) models. However, for the TVZIP-INGARCH (1, 1) case where the GARCH portion of the parameters ($\alpha_1, \beta_1$) show relatively less accurate estimates, for both procedures even for the large sample size case. For instance, when $N = 120$ with true values of $\alpha_1 = 0.40$ and $\beta_1 = 0.30$ the means of the corresponding parameter estimates are 0.3916 and 0.2439 respectively, for the MLE method, and when $N = 360$ these means of estimates change to 0.4794 and 0.2205 respectively (see Table 9, Model C3). A similar phenomenon is seen in the simulation results of Zhu [2]. As done in Case 1, using additional initial value combinations did not yield any improvement in this regard. In general, MADE values decrease with the increase in sample size.

**Table 8.** Means of MLE Estimates, MADE (within parentheses), and computational efforts (CPU time) for TVZIP-INARCH (2) models where zero-inflation is driven by an exogenous variable.

| Model | $N$ | $\hat{\delta}_0$ | $\hat{\delta}_1$ | $\hat{\alpha}_0$ | $\hat{\alpha}_1$ | $\hat{\alpha}_2$ | CPU time (seconds) |
|---|---|---|---|---|---|---|---|
| | True values | -2.00 | 0.00 | 1.00 | 0.20 | 0.20 | |
| B1 | 120 | -2.4576 (0.7430) | -0.0399 (0.5504) | 1.0217 (0.1560) | 0.1947 (0.0741) | 0.1846 (0.0729) | 43.27 |
| | 360 | -2.1479 (0.3557) | 0.0024 (0.2779) | 1.0147 (0.0950) | 0.1956 (0.0474) | 0.1904 (0.0453) | 176.73 |
| | True values | -1.00 | -1.00 | 2.00 | 0.30 | 0.20 | |
| B2 | 120 | -1.0505 (0.2390) | -1.0593 (0.2436) | 2.0354 (0.2494) | 0.2911 (0.0829) | 0.1930 (0.0721) | 39.06 |
| | 360 | -1.0132 (0.1258) | -1.0191 (0.1345) | 2.0092 (0.1418) | 0.2989 (0.0476) | 0.1977 (0.0453) | 177.13 |
| | True values | 2.00 | 1.00 | 1.00 | 0.40 | 0.30 | |
| B3 | 120 | 1.9816 (0.4490) | 1.2290 (0.4727) | 0.9402 (0.2888) | 0.3128 (0.2754) | 0.2651 (0.2250) | 53.70 |
| | 360 | 1.9801 (0.2139) | 1.0532 (0.1973) | 0.9797 (0.1665) | 0.3586 (0.2286) | 0.2953 (0.1941) | 208.41 |

**Table 9.** Means of MLE Estimates, MADE (within parentheses), and computational efforts (CPU time) for TVZIP-INGARCH (1, 1) models where zero-inflation is driven by an exogenous variable.

| Model | N | $\hat{\delta}_0$ | $\hat{\delta}_1$ | $\hat{\alpha}_0$ | $\hat{\alpha}_1$ | $\hat{\beta}_1$ | CPU time (seconds) |
|---|---|---|---|---|---|---|---|
| | True values | -2.00 | 0.00 | 1.00 | 0.20 | 0.20 | |
| C1 | 120 | -2.4440 (0.7375) | -0.0414 (0.5675) | 0.9240 (0.2168) | 0.2247 (0.0787) | 0.2132 (0.1421) | 698.89 |
| | 360 | -2.1381 (0.3509) | 0.0014 (0.2714) | 0.9397 (0.1527) | 0.2278 (0.0511) | 0.1889 (0.1071) | 2417.10 |
| | True values | -1.00 | -1.00 | 2.00 | 0.30 | 0.20 | |
| C2 | 120 | -1.0412 (0.2390) | -1.0651 (0.2595) | 1.7454 (0.3311) | 0.3928 (0.1144) | 0.1586 (0.0982) | 393.83 |
| | 360 | -1.0088 (0.1249) | -1.0137 (0.1346) | 1.8083 (0.2164) | 0.3992 (0.1021) | 0.1337 (0.0869) | 1111.40 |
| | True values | 2.00 | 1.00 | 1.00 | 0.40 | 0.30 | |
| C3 | 120 | 2.0149 (0.4211) | 1.1580 (0.4189) | 0.8308 (0.3308) | 0.3916 (0.2999) | 0.2439 (0.2477) | 762.32 |
| | 360 | 1.9965 (0.2223) | 1.0465 (0.2018) | 0.8229 (0.2391) | 0.4794 (0.2650) | 0.2205 (0.2199) | 4092.20 |

## Extension to the softplus INGARCH model

Herein we introduce a generalization of the softplus INGARCH model proposed by Weiß et al. [12] by incorporating zero-inflation to the underlying count-data distribution and allowing a set of exogenous variables to act as inputs to the model that defines the conditional mean of the process. In the following, we assume that the underlying distribution is zero-inflated versions of the Poisson or the negative binomial distributions or the generalized Poisson. The zero-inflated softplus INGARCHX model we developed is defined as follows.

Let $\{X_t : t \in \mathbb{Z}\}$ is the count process, and $\mu_t$ defines the conditional mean of $X_t$ given the sigma-fields $\mathbb{F}_{t-1}^X$ and $\mathbb{F}_t^V$ such that $\mu_t = E(X_t | \mathbb{F}_{t-1}^X, \mathbb{F}_t^V, \Theta)$. Here, $\mathbb{F}_{t-1}^X$ is the sigma-field generated by $\{X_l, \lambda_l; l \leq t-1\}$ and $\mathbb{F}_t^V$ is the sigma-field generated by the set of exogenous covariates $\{\mathbf{V_1}, \ldots, \mathbf{V_t}\}$ with $\mathbf{V}_j = (V_{j1}, V_{j2}, \ldots, V_{jr})^T$ for $j = 1, \ldots, t$. Note that $\Theta$ is a set of parameters defined later in the section.

Let $q(.|\lambda_t^*, \Omega)$ be a probability mass function of a discrete random variable $U_t$ $U_t$, with conditional $\lambda_t^*$, for $t \in \mathbb{Z}$. We define $\lambda_t$ such that $\lambda_t = \lambda_t^*$ if the distribution of $U_t$ is Poisson or negative binomial, and $\lambda_t = (1 - \varphi)\lambda_t^*$ if the distribution is generalized Poisson, where $\varphi$ is the dispersion parameter. This defines the underlying count data distribution without zero inflation. Now let $\omega$ be the zero-inflation probability associated with $X_t$. Then the conditional probability mass function of $X_t | \mathbb{F}_{t-1}^X, \mathbb{F}_t^V, \Theta$ is given by,

**Table 10.** Means of EM Estimates, MADE (within parentheses), and computational efforts (CPU time) for TVZIP-INARCH (1) models where zero-inflation is driven by an exogenous variable.

| Model | N | $\hat{\delta}_0$ | $\hat{\delta}_1$ | $\hat{\alpha}_0$ | $\hat{\alpha}_1$ | CPU time (seconds) |
|---|---|---|---|---|---|---|
| | True values | -2.00 | 0.00 | 1.00 | 0.40 | |
| A1 | 120 | -2.4239 (0.7071) | -0.0326 (0.5467) | 1.0171 (0.1382) | 0.3808 (0.0836) | 702.16 |
| | 360 | -2.0951 (0.3226) | -0.0156 (0.2538) | 1.0144 (0.0816) | 0.3892 (0.0487) | 4101.90 |
| | True values | -1.00 | -1.00 | 2.00 | 0.50 | |
| A2 | 120 | -1.0469 (0.2384) | -1.0714 (0.2583) | 2.0193 (0.2143) | 0.4864 (0.0836) | 212.59 |
| | 360 | -1.0209 (0.1338) | -1.0182 (0.1400) | 2.0073 (0.1198) | 0.4932 (0.0476) | 1198.40 |
| | True values | 2.00 | 1.00 | 1.00 | 0.70 | |
| A3 | 120 | 2.0081 (0.4317) | 1.1600 (0.4141) | 0.9915 (0.2973) | 0.4607 (0.3699) | 506.34 |
| | 360 | 1.9975 (0.2197) | 1.0365 (0.1947) | 1.0037 (0.1694) | 0.5662 (0.2739) | 2587.20 |

**Table 11.** Means of EM Estimates, MADE (within parentheses), and computational efforts (CPU time) for TVZIP-INARCH (2) models where zero-inflation is driven by an exogenous variable.

| Model | $N$ | $\hat{\delta}_0$ | $\hat{\delta}_1$ | $\hat{\alpha}_0$ | $\hat{\alpha}_1$ | $\hat{\alpha}_2$ | CPU time (seconds) |
|---|---|---|---|---|---|---|---|
|  | True values | -2.00 | 0.00 | 1.00 | 0.20 | 0.20 |  |
| B1 | 120 | -2.4495 (0.7348) | -0.0363 (0.5440) | 1.0218 (0.1561) | 0.1947 (0.0741) | 0.1847 (0.0729) | 869.83 |
|  | 360 | -2.1466 (0.3543) | 0.0032 (0.2769) | 1.0148 (0.0950) | 0.1956 (0.0474) | 0.1905 (0.0453) | 5429.10 |
|  | True values | -1.00 | -1.00 | 2.00 | 0.30 | 0.20 |  |
| B2 | 120 | -1.0505 (0.2390) | -1.0593 (0.2436) | 2.0354 (0.2494) | 0.2911 (0.0829) | 0.1930 (0.0721) | 256.73 |
|  | 360 | -1.0132 (0.1258) | -1.0191 (0.1345) | 2.0092 (0.1418) | 0.2989 (0.0476) | 0.1977 (0.0453) | 1469.20 |
|  | True values | 2.00 | 1.00 | 1.00 | 0.40 | 0.30 |  |
| B3 | 120 | 1.9903 (0.4404) | 1.2145 (0.4581) | 0.9405 (0.2885) | 0.3140 (0.2752) | 0.2640 (0.2243) | 656.63 |
|  | 360 | 1.9800 (0.2139) | 1.0532 (0.1973) | 0.9796 (0.1655) | 0.3586 (0.2286) | 0.2945 (0.1937) | 3599.00 |

$$P(X_t = k | \mathbb{F}_{t-1}^X, \mathbb{F}_t^V, \boldsymbol{\Theta}) = \omega \delta_{k,0} + (1-\omega) q(X_t = k | \lambda_t, \boldsymbol{\Omega}), \text{ where } \delta_{k,0} = \begin{cases} 1; k = 0 \\ 0; k \neq 0 \end{cases}. \text{ Then}$$

the conditional mean of $X_t$ given $\mathbb{F}_{t-1}^X$ and $\mathbb{F}_t^V$ is $\mu_t = E(X_t | \mathbb{F}_{t-1}^X, \mathbb{F}_t^V, \boldsymbol{\Theta}) = (1-\omega) g(\lambda_t)$, for $t \in \mathbb{Z}$. If $q(.|\lambda_t^*, \boldsymbol{\Omega})$ is representing a Poisson, negative binomial Type 1 or negative binomial Type 2 distribution, then the $g(\lambda_t) = \lambda_t$. If the underlying distribution is generalized Poisson, then $g(\lambda_t) = \frac{\lambda_t}{1-\varphi}$. If the proposed softplus INGARCHX version, we let $\lambda_t$ to be defined as

$$\lambda_t = s_c\left(\alpha_0 + \sum_{i=1}^p \alpha_i X_{t-i} + \sum_{j=1}^q \beta_j \lambda_{t-1} + \sum_{k=1}^r \gamma_{tk} V_{tk}\right),$$

with the softplus link function given by

$s_c(z) = c \ln\left(1 + \exp\left(\frac{z}{c}\right)\right)$ with $c > 0$.

The set of parameters in the model can now be defined as

$\boldsymbol{\Theta} = (\omega, \alpha_0, \alpha_1, \ldots, \alpha_p, \beta_1, \ldots, \beta_q, \gamma_{11}, \ldots, \gamma_{tr}, c, \boldsymbol{\Omega})$.

## Real data examples

In this section the proposed TVZIP-INGARCH $(p, q)$ model is applied to two real-world data-sets. The time-varying component given by Eq (5) can be formulated in many ways. In example one, we selected two scenarios for this component, and labeled them Sc1 and Sc2. In example two, the time-varying component was labeled as obeying Scenario 4 (Sc4). Details

**Table 12.** Means of EM Estimates, MADE (within parentheses), and computational efforts for TVZIP-INGARCH (1, 1) models where zero-inflation is driven by an exogenous variable.

| Model | $N$ | $\hat{\delta}_0$ | $\hat{\delta}_1$ | $\hat{\alpha}_0$ | $\hat{\alpha}_1$ | $\hat{\beta}_1$ | CPU time (seconds) |
|---|---|---|---|---|---|---|---|
|  | True values | -2.00 | 0.00 | 1.00 | 0.20 | 0.20 |  |
| C1 | 120 | -2.3593 (0.7318) | -0.0507 (0.5409) | 0.9171 (0.2169) | 0.2241 (0.0791) | 0.2002 (0.1257) | 1167.80 |
|  | 360 | -2.1129 (0.3541) | 0.0007 (0.2634) | 0.9389 (0.1504) | 0.2274 (0.0515) | 0.1860 (0.1020) | 6230.40 |
|  | True values | -1.00 | -1.00 | 2.00 | 0.30 | 0.20 |  |
| C2 | 120 | -1.0405 (0.2391) | -1.0645 (0.2591) | 1.7403 (0.3341) | 0.3926 (0.1145) | 0.1595 (0.0979) | 415.02 |
|  | 360 | -1.0088 (0.1249) | -1.0137 (0.1346) | 1.8083 (0.2164) | 0.3992 (0.1021) | 0.1337 (0.0869) | 1457.60 |
|  | True values | 2.00 | 1.00 | 1.00 | 0.40 | 0.30 |  |
| C3 | 120 | 2.0226 (0.4137) | 1.1355 (0.4003) | 0.8078 (0.3335) | 0.3950 (0.2991) | 0.2325 (0.2371) | 1019.20 |
|  | 360 | 1.9917 (0.2254) | 1.0414 (0.2005) | 0.7981 (0.2574) | 0.4808 (0.2640) | 0.2177 (0.2117) | 4885.10 |

regarding these scenarios can be found under each example. The proposed model's performance is compared with constant zero-inflated probability versions of the INGARCH $(p, q)$ model such as zero-inflated Poisson (ZIP), zero-inflated negative binomial Type 1(ZINB1), zero-inflated negative binomial Type 2 (ZINB2), all proposed by Zhu [2], and the zero-inflated generalized Poisson (ZIGP) model proposed by Chen and Lee [18]. The zero-inflated compound Poisson INGARCH models such as zero-inflated geometric Poisson INGARCH (ZIGEOMP-INGARCH) and the zero-inflated Neyman Type A INGARCH (ZINTA-IN-GARCH) of Gonçalves et al. [21], were also fitted to the data. In addition, the following formulations were also included in the comparison. The log-linear INGARCH model of Fokianos and Tjøstheim [10] was modified to include zero-inflation. We also modified the log-linear INGARCHX model of Chen and Lee [11] to include zero-inflation, but it should be noted that with the inclusion of zero-inflation the above two models are nested within the non-time-varying version of the model of Xu et al. [30]. We extended the softplus INGARCH model introduced by Weiß et al. [12], as described in an earlier section, by introducing zero inflation and an exogenous variable. Apart from our proposed models, this provides us with six distinct model categories or model structures, namely the zero-inflated versions of C1: INGARCH, C2: compound Poisson INGARCH, C3: log-linear INGARCH, C4: log-linear INGARCHX, C5: softplus INGARCH, and C6: softplus INGARCHX. We denote the versions of our proposed model under the C0 category. Note that model categories C1 –C6 all have constant zero-inflation probability, and hence for consistency, we label them as falling under zero-inflation Scenario 3 (Sc3).

The INGARCH, log-linear INGARCH, log-linear INGARCHX, the softplus INGARCH, and the softplus INGARCHX models were fitted assuming the zero-inflated versions of the Poisson, negative binomial Type 1, negative binomial Type 2, and generalized Poisson distributions. For definitions of negative binomial Type 1 and Type 2 distributions, the reader is referred to Zhu [2]. Note that the compound Poisson model of Gonçalves et al. [21] was fitted assuming the zero-inflated geometric Poisson and the zero-inflated Neyman Type A distributions. The above combinations of model categories and distributions yield a total of twenty-three count data time series formulations. These twenty-three formulations are listed in Table 1 in S4 Appendix. These formulations were then fitted using three model orders, namely M1 with $p = 1, q = 0$; M2 with $p = 2, q = 0$; and M3 with $p = 1, q = 1$. This gives rise to sixty-nine total models.

Not all of these sixty-nine models were fitted to data in each of the two examples shown below. Example 1 does not contain data on an exogenous variable and hence the log-linear INGARCHX and the softplus INGARCHX model categories were not used in the first example. Models belonging to the above two categories, however, were fitted to the data in Example 2 in place of the log-linear INGARCH and the softplus INGARCH model categories. Excluding versions of the proposed model, forty-five models with constant zero-inflation probability were fitted to each of the example data sets, with eighteen models common to both sets (twelve models from C1 category and six models from C2 category).

Since we are comparing the proposed time-varying zero inflation model against models with constant zero-inflation probability, an argument can be made that any superior performance by the proposed model can be due to the presence of a change point. Thus, fitting a suitably selected constant zero-inflation model separately to the two sub-series before and after a detected change point may negate any such superiority. To address this potential criticism, the following approach was taken with respect to the INGARCH models (C1 category) with zero-inflated versions of Poisson, negative binomial Type 1, negative binomial Type 2, and the generalized Poisson distributions under model orders M1, M2, and M3. First, we fitted the above set of constant zero-inflation models to the example data. Then the model with the smallest

AIC value from among the zero-inflated INGARCH models within each model order M1, M2, M3 was selected. We repeated the same procedure using the Bayesian Information Criteria (BIC) and found that the models selected using the AIC values also had the lowest BIC values. Following that, the CUSUM test proposed by Lee et al. [27] was applied using the residuals from the selected model. If a change point was detected, the model under consideration was fitted independently to the two subseries before and after the change point. It is noteworthy that a change point was detected every time this test was carried out. The AIC and BIC values were computed for all models, including in cases where the model was fitted to two sub-series. Note that the theoretical properties of the CUSUM test proposed by Lee et al. [27] are based on ten assumptions which holds for the zero-inflated versions of Poisson, negative binomial, and generalized Poisson, according to Lee et al. [27] and Lee and Lee [32]. We did not, however, verify whether models in other categories would also satisfy all these assumptions. Thus, we carried out this procedure for all models falling into the zero-inflated INGARCH (C1) category but did not do so for other model categories.

Following the above process, we identified models with the lowest AIC and BIC values within each model category (C1, C2, C3, C4, C5, C6) by model order (M1, M2, M3) combination. Models thus identified can be seen in cells highlighted in light gray in Table 1B of S5 Appendix (for Example 1) and Table 1B of S6 Appendix (for Example 2). Note that Table 1A in each Appendix provides the AIC and BIC values for versions of our proposed model. Within each model category (but across model orders), we identified the model with the lowest AIC value and the model with the lowest BIC value among models highlighted in light gray. Then from among the above models, we also identified the best model(s) across all model categories by using AIC and BIC values. After the best model or models across all model categories were selected, standardized Pearson residuals were calculated, and used to check model adequacy.

The first example is based on the *Influenza A associated pediatric deaths* data set downloaded from the Centre for Disease Control and Prevention web page. This data was modelled by using the TVZIP-INGARCH ($p$, $q$) formulation with a sinusoidal zero-inflation function. In the second example, we used the *Pediatric mortality caused by Influenza B* data set downloaded from the same web page to demonstrate the performance of TVZIP-INGARCH ($p$, $q$) process where the zero-inflation is driven by an exogenous variable. The data sets used in this study are available in the supplement labeled S1 Data.

## Real data example—Use of a deterministic sinusoidal zero-inflation function

The TVZIP-INARCH (1), TVZIP-INARCH (2), and TVZIP-INGARCH (1, 1) models with sinusoidal zero-inflation were fitted to the *Influenza A associated pediatric mortality* data set. Performance of the proposed models was compared to those of the constant zero-inflated count data time series models mentioned above.

The data set provides weekly count data of U.S. pediatric deaths caused by type A influenza virus over the time period from week 40 of 2015 to week 43 of year 2018. This consists of 160 weekly observations of pediatric death counts. The data were extracted from weekly U.S. Influenza Surveillance report, which was published by Centre for Disease Control and Prevention (CDC) [33]. Summary statistics of the data showed a mean of 1.5063 and a variance of 6.352, suggestive of overdispersion. Fig 1 illustrates the frequency of each pediatric mortality case caused by influenza A virus using a bar chart. Observe that there are 86 zeros, which comprises 53.75% of the total time points.

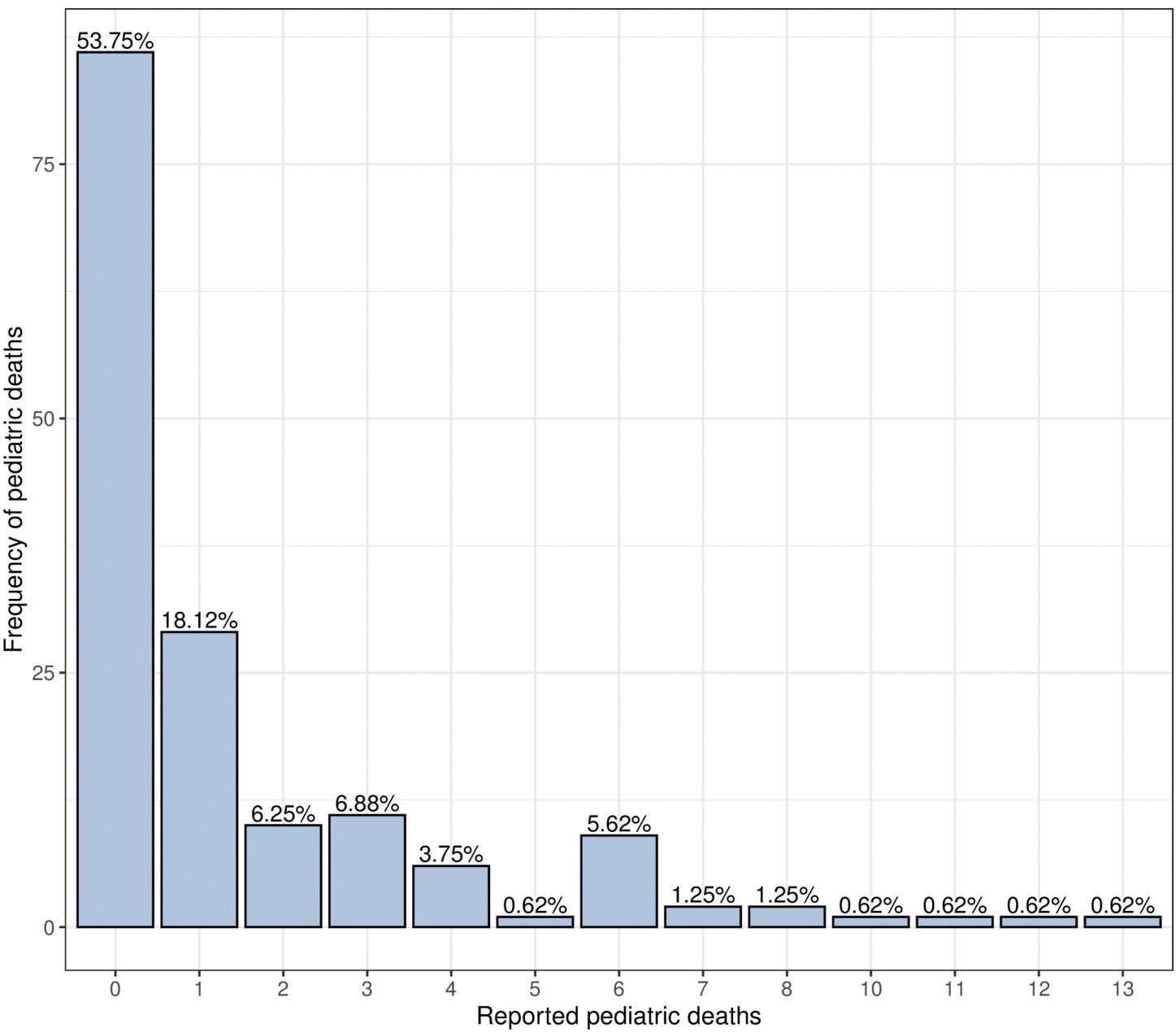

**Fig 1. Bar chart of Influenza A associated pediatric deaths.**

Fig 2 illustrates the original time series of the counts followed by the plots of its autocorrelation function (ACF) and the partial autocorrelation function (PACF), respectively. The bar chart shows the excess number of zeros in the data, and the time series plot demonstrates prolonged periods of zero counts, which supports the use of zero-inflated Poisson time series models to analyze this data. Furthermore, we can observe an annual seasonality in the portions of the time series with excessive zero counts.

To understand the zero-inflation behaviour of this data set, we aggregated weekly data in to its corresponding calendar month and constructed the total monthly zero mortality counts. These counts were then converted into a monthly proportion by dividing each monthly total

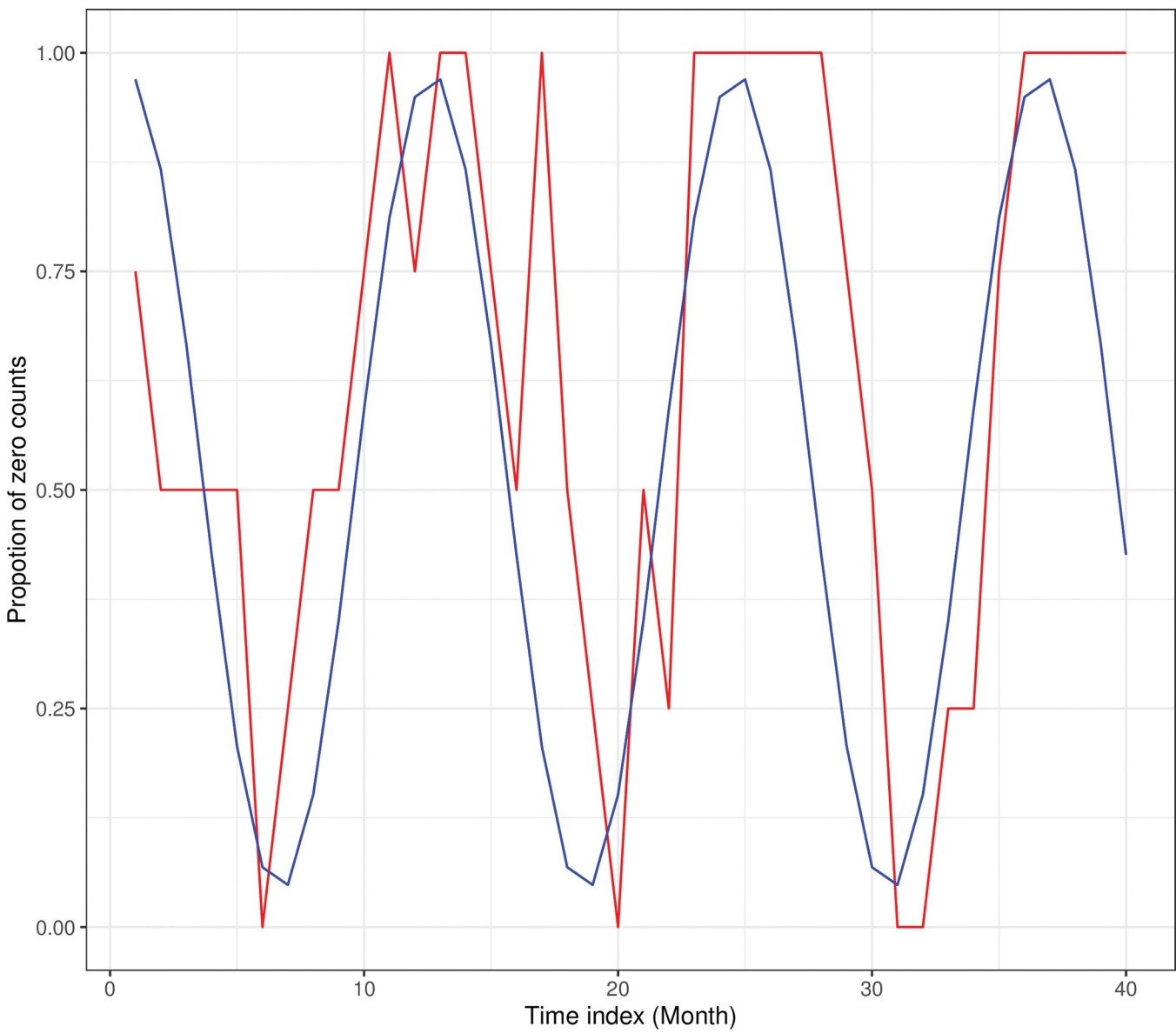

**Fig 2. Influenza A associated pediatric deaths, sample ACF plot, and sample PACF plot.**

zeros by the maximum zero count over the observed period to scale values at or below one. According to the Fig 3, the plot of the monthly proportion of zero counts exhibits an approximate sinosidual behaviour throughout the observed time span. Thus, the general sinosidual function mentioned in Eq (5) can be used to model the zero-inflation behaviour of this data. More details of this modelling will be described later in this section.

As mentioned earlier, two scenarios were considered for the time-varying zero-inflation component of the proposed model. In the models listed under Scenario 1 (Sc1) we assumed a piecewise constant zero-inflation fucntion. The monthly zero-inflation was allowed to vary according to a sinusoidal zero-inflation function with a 12-month period, but the weeks within a given month were assigned the same zero-inflated probability value associated with that month. The models under Scenario 2 (Sc2) used a sinusoidal function describing a time-

## Time series plot of Influenza A associated pediatric deaths

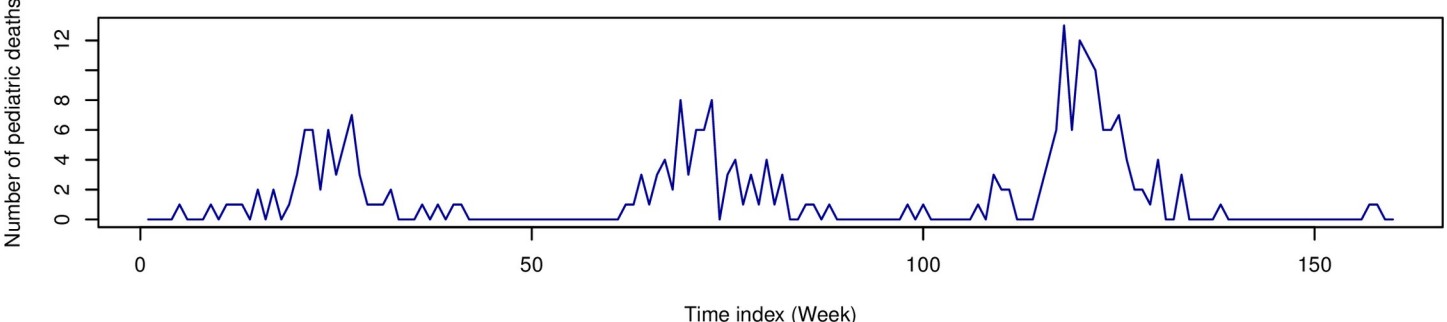

## ACF Plot

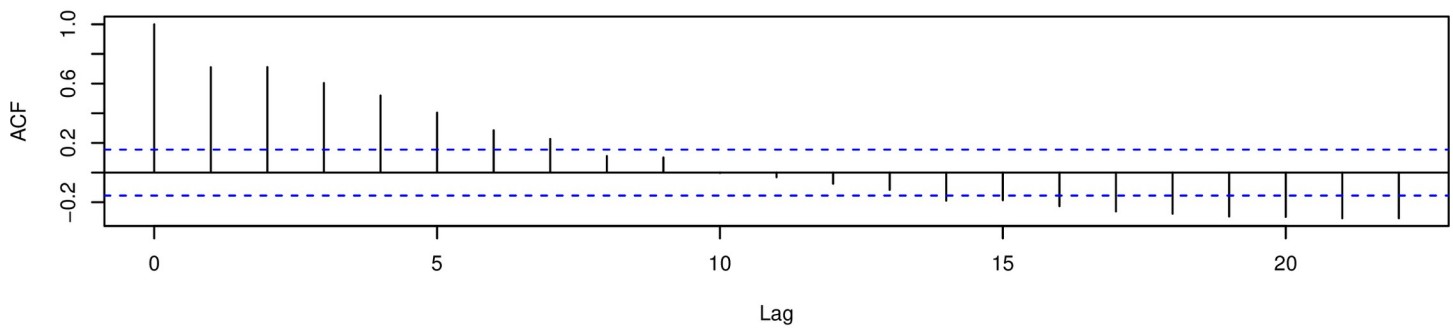

## PACF Plot

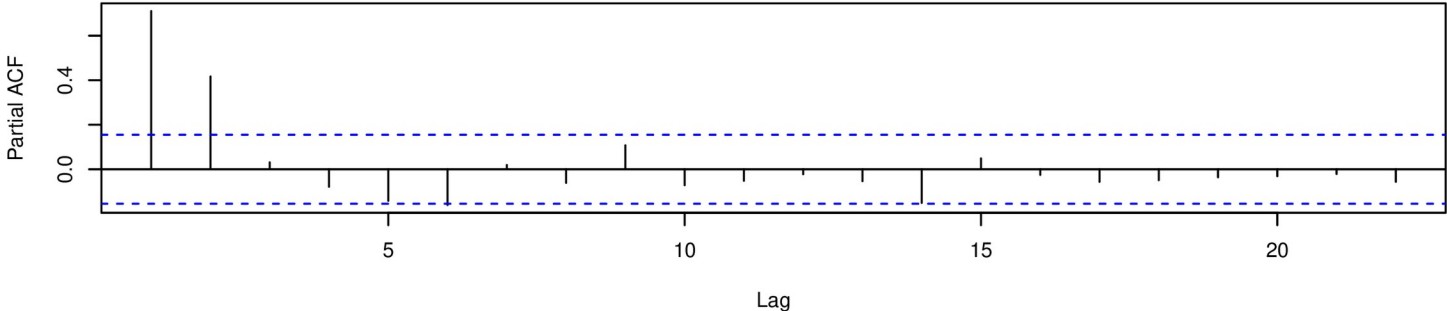

**Fig 3. Monthly proportion of zero mortality counts versus the fitted sinusoidal zero-inflated function.** (Red) Monthly proportion of zero mortality counts. (Blue) Fitted sinusoidal zero-inflated function.

varying zero-inflation probability that changes on a weekly basis. Note that all other model categories we consider fall under Scenario 3 (Sc3) where the zero-inflation probability remains constant over time. In addition to the proposed model under scenarios Sc1, and Sc2, we fitted models in categories C1, C2, C3, and C5. All models were fitted under the model orders M1, M2, M3 with various underlying zero-inflated distributions described at the beginning of the real data example section. The EM algorithem was used to estimate the model parameters and the results for Sc1 and Sc2 scenarios are reported in the Table 13. The EM algorithm was chosen over the MLE method to be in line with the method used in Zhu [2] for its real data example.

**Table 13.** Estimated parameters, standard errors (within parentheses), AIC and BIC for the pediatric death counts caused by virus A.

| Model | Estimated parameters and standard errors | | | | | | Model selection criteria | |
|---|---|---|---|---|---|---|---|---|
| | $\hat{A}$ | $\hat{B}$ | $\hat{\alpha}_0$ | $\hat{\alpha}_1$ | $\hat{\alpha}_2$ | $\hat{\beta}_1$ | AIC | BIC |
| Sc1 C0 M1 | 0.1247 (0.0465) | 0.4139 (0.0284) | 1.1228 (0.1767) | 0.7148 (0.0805) | | | 411.8966 | 424.1973 |
| **Sc1 C0 M2** | 0.0999 (0.0470) | 0.3956 (0.0300) | 0.7661 (0.1578) | 0.4720 (0.0993) | 0.3564 (0.0957) | | **398.7016** | **414.0774** |
| Sc1 C0 M3 | 0.0961 (0.1913) | 0.3917 (0.0737) | 0.3897 (0.6358) | 0.4839 (0.3399) | | 0.4378 (0.3848) | 402.5728 | 417.9487 |
| Sc2 C0 M1 | -0.2700 (0.0416) | 0.3191 (0.0373) | 1.0622 (0.1679) | 0.7194 (0.0793) | | | 419.6800 | 431.9807 |
| Sc2 C0 M2 | -0.2694 (0.0404) | 0.2704 (0.0402) | 0.6759 (0.1435) | 0.4587 (0.0961) | 0.3847 (0.0938) | | 403.6973 | 419.0732 |
| Sc2 C0 M3 | -0.2643 (0.0402) | 0.2550 (0.0404) | 0.2987 (0.1800) | 0.4739 (0.1001) | | 0.4682 (0.1231) | 408.1727 | 423.5486 |

Table 13 shows that the models which fall under the Sc1 exhibited lower AIC and BIC values compared to their counterparts under Sc2. We could conclude that the use of a piecewise constant time-varying zero-inflation function improved the fit compared to version where the time-varying zero-inflation probability changes on a weekly basis.

As described ealier, both the AIC and BIC values were used to identify the best models within each model category by model order combination for the rest of the constant zero-inflation models. The models thus identified are shown in light gray highlighted cell in Tables 1A and 1B in S5 Appendix. Models with the lowest AIC or BIC values within each model category are identifies in boldface type in Tables 1A and 1B in S5 Appendix. If the model has the lowest information criteria values with respect to both AIC and BIC, then it is identified in italic boldface font. These models with the lowest AIC or BIC within each model category are also shown listed in Table 14, togther with the Sc1 and Sc2 versions of the proposed model under model order M2, which was selected as the best among all three orders.

The performance of the two versions of the TVZIP-INGARCH ($p$, $q$) models with the lowest AIC and BIC values are compared with those of other INGARCH time series models that assume constant zero-inflation in Table 14. Among all the models, the TVZIP-INARCH (2) under Sc1 zero-inflation assumption has lower AIC and BIC values. Note that the second best model with respect to AIC and BIC criteria is also a version of the proposed TVZIP-IN-GARCH model.

The TVZIP-INARCH (2) model, with the piecewise zero-inflation formulation, which has the lowest AIC and BIC values amoung all models considered, was selected for further investigation. The standardized Pearson residuals were calculated for the selected model and used for diagnostic tests. The ACF plot and the histogram, both based on the standardized Pearson residuals are presented in Fig 4.

**Table 14.** The AIC and BIC values for the best model in model catergory C1,C2, C3 and C5 under Scenario 3 and the best model order for the proposed model under Scenarios Sc1 and Sc2.

| Zero-inflated scenario | Model information | | Model selection criteria | |
|---|---|---|---|---|
| | Model category | Model Order | AIC | BIC |
| Sc1 | C0: Time-varying zero-inflated Poisson INGARCH | *TVZIP-INARCH (2)* | *398.7016* | *414.0774* |
| Sc2 | C0: Time-varying zero-inflated Poisson INGARCH | TVZIP-INARCH (2) | 403.6973 | 419.0732 |
| Sc3 | C1: Zero-inflated INGARCH | ZINB1-INARCH (2) | 412.3557 | 427.6686 |
| | | ZINB1-INARCH (2)—change point | 410.2882 | 433.6915 |
| | C2: Zero-inflated compound Poisson INGARCH | ZIGEOMP-INARCH (2) | 412.4334 | 427.7463 |
| | C3: Zero-inflated log-linear INGARCH | ZINB1 log-linear INARCH (2) | 418.3061 | 433.6191 |
| | C5: Zero-inflated softplus INGARCH | ZINB1 softplus INARCH (2) | 416.8903 | 432.2032 |

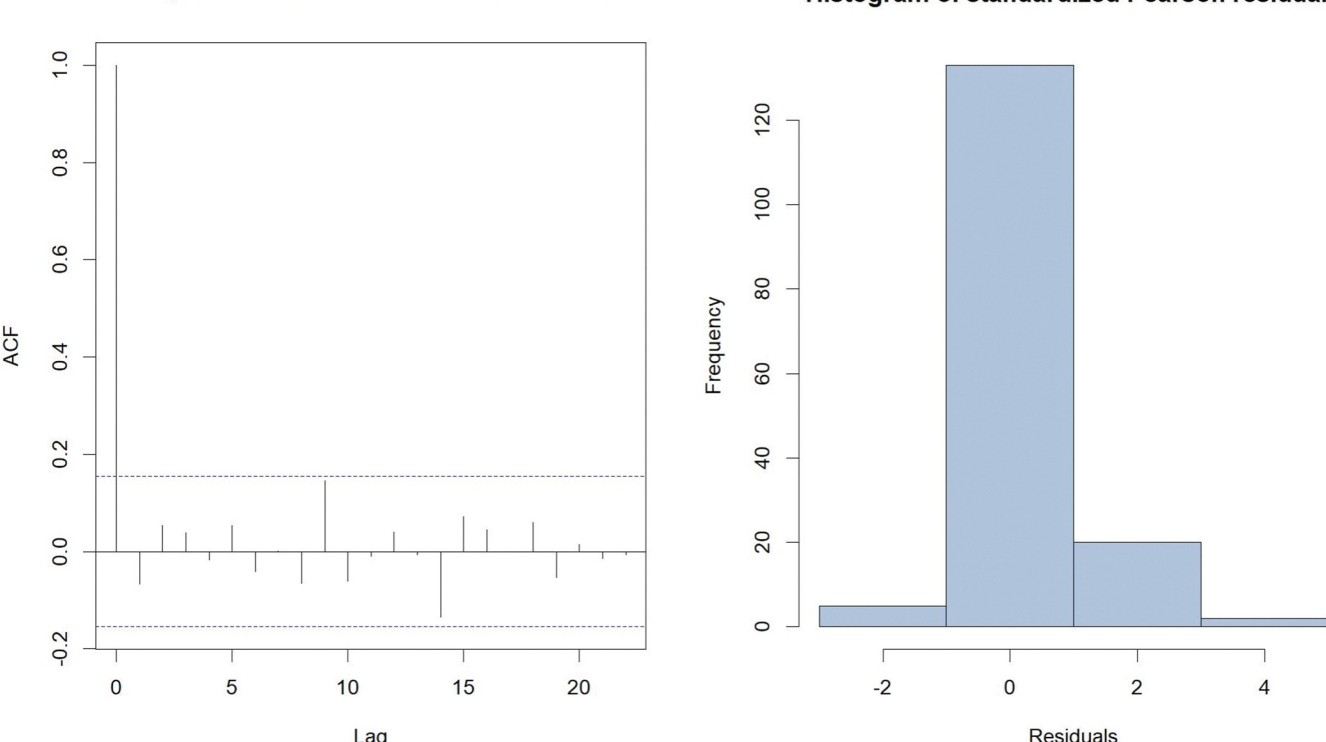

**Fig 4. The ACF plot and the histogram, both based on the standardized Pearson residuals of fitted TVZIP-INARCH (2) model.**

The standardized Pearson residuals display a mean (0.0017), which is close to zero, and a variance (0.8108), which is close to one. The ACF plot indicates that residuals of the fitted model meet the assumption of no autocorrelation at 5% significance level. Additionally, the mean of the fitted model (1.4608) is close to the observed mean of the count data series (1.5063). Based on results from both model selection criteria, model fit, and residual diagnostic results, we conclude that the TVZIP-INARCH (2) model with Sc1 formulation for zero-inflation probability provided the best fit to the data. In this model, we assumed that weeks within any given month has a constant zero-inflation, yet monthly zero-inflation varies cyclically.

### Real data example—Zero-inflation function is driven by exogenous variable

In this subsection we examine the performance of the TVZIP-INGARCH $(p, q)$ models when the zero-inflation function is driven by an exogenous variable. *Influenza B associated pediatric mortality* data set was used, and the TVZIP-INARCH (1), TVZIP-INARCH (2), and TVZIP-INGARCH (1, 1) models were fitted to the data. Results were compared with forty-five constant zero-inflated INGARCH $(p, q)$ models. We selected the weekly average of nationwide low temperatures as the exogenous variable that drives the zero-inflation probability. This is motivated by one of the reasons generally accepted as a cause for the easy transmission of influenza during winter months, namely the cold temperatures driving people indoors. Influenza B associated pediatric mortality data set was accessed from the weekly U.S. Influenza Surveillance Report [33], and the temperature data were extracted from the United States National Oceanic and Atmospheric Administration (NOAA) weather prediction center [34]. Both data sets spanned the time period from week 40 in year 2014 to week 39 in year 2018.

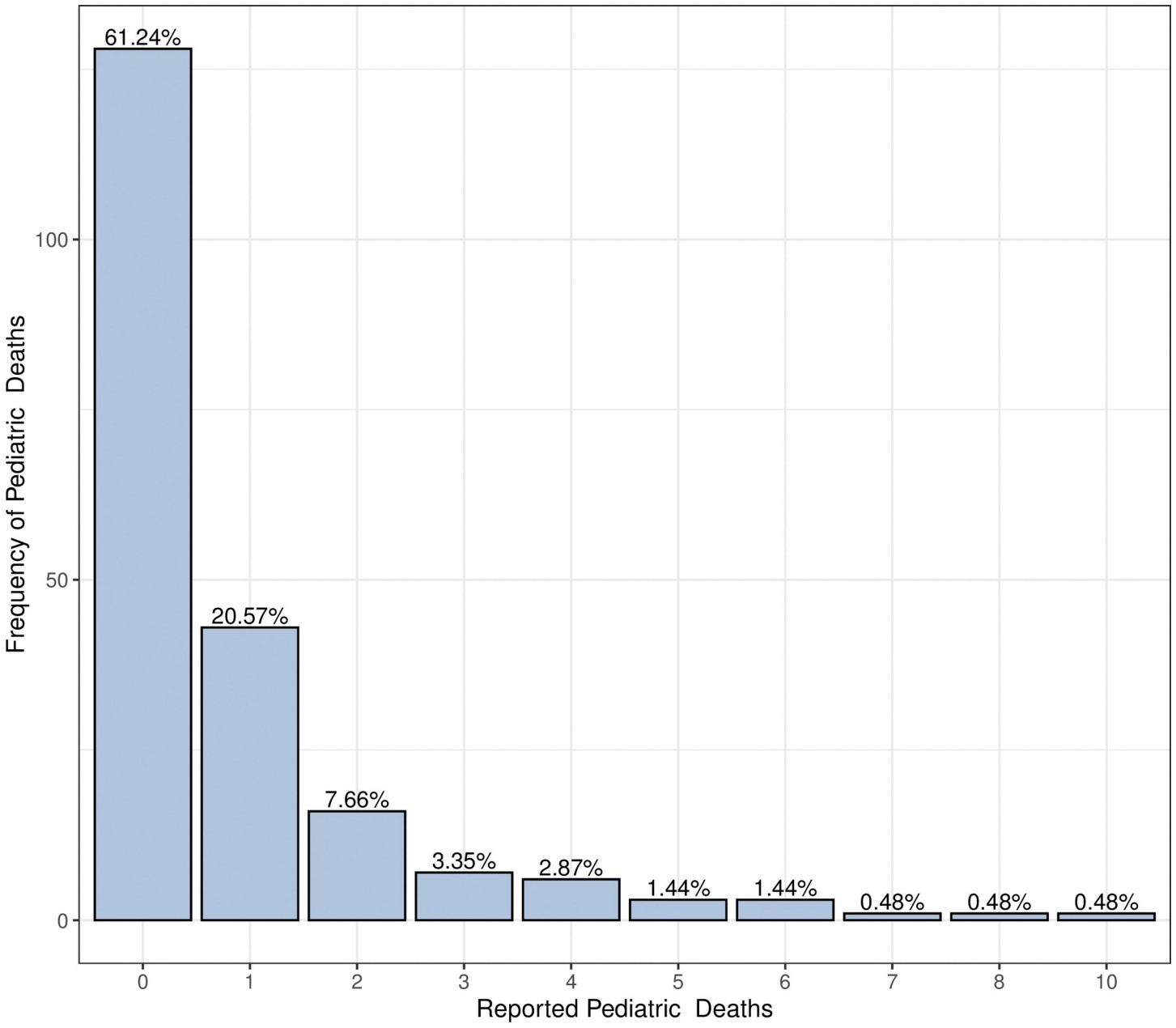

**Fig 5. Bar chart of influenza B associated pediatric deaths.**

The influenza B data set contained 209 weekly observations of pediatric death counts. Summary statistics of pediatric mortality cases due to influenza B showed a mean of 0.8517 and a variance of 2.4538. Since the empirical variance was higher than the empirical mean, the data exhibit overdispersion. Fig 5 illustrates the frequency of each pediatric mortality count caused by influenza B virus type using a bar chart. The bar chart shows that there are 128 zeros, which comprises 61.24% of total of the time points. This suggests that the pediatric mortality data are zero-inflated.

The time series plot, ACF, and PACF plots are given in Fig 6. Based on time series plot, we can see that there is an annual seasonality exhibited in this data set. Moreover, it shows that

**Time Series Plot of Influenza B Associated Pediatric Deaths**

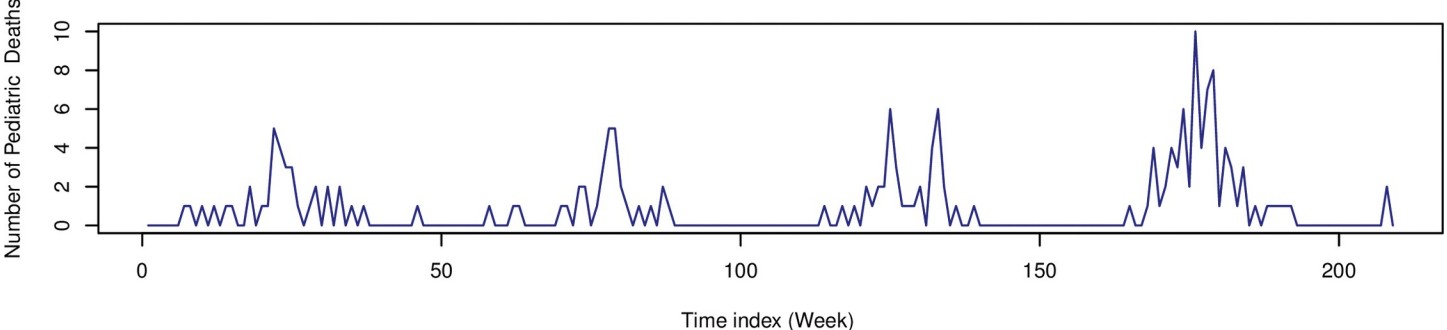

**ACF plot**

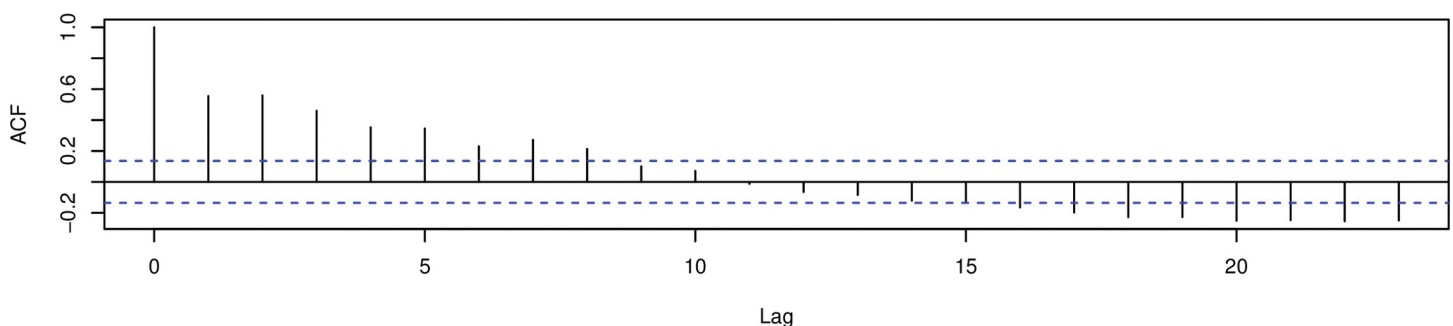

**PACF plot**

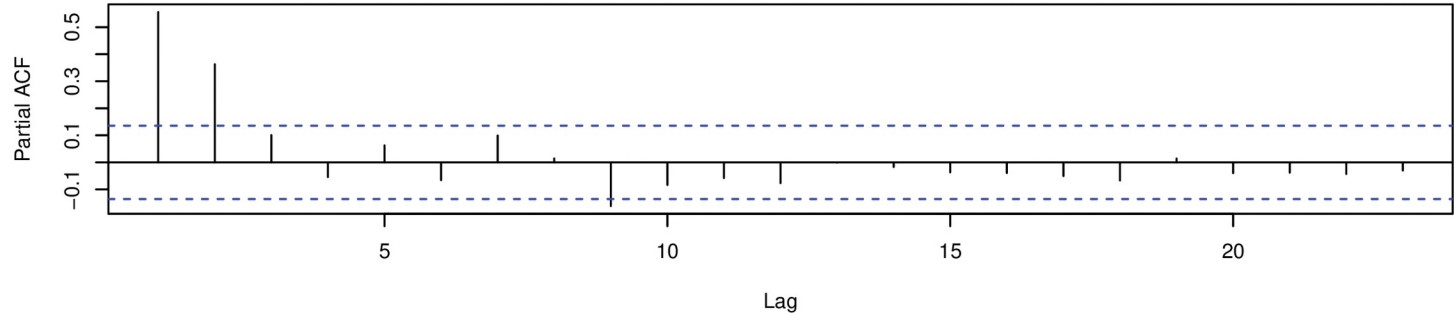

**Fig 6. Influenza B associated pediatric deaths, sample ACF plot, and sample PACF plot.**

there were periods with clusters of zeros that occur periodically. Therefore, as discussed in the above section, we suggested TVZIP-INGARCH ($p$, $q$) to model the count data series.

In this example, the time-varying zero-inflation was modelled by considering an exogenous time series, namely the weekly average of nationwide low temperature. Comparison of the two time series plots is given in Fig 7.

Fig 7 shows that periods with higher values of low temperature coincide with periods of zero pediatric mortality caused by Influenza B. Hence, it is suggestive that periods with high zero counts (low pediatric mortality) are related to periods with higher values of low temperature. Thus, we used the weekly average low temperature data to model the time-varying zero-

## Influenza B Associated Pediatric Deaths

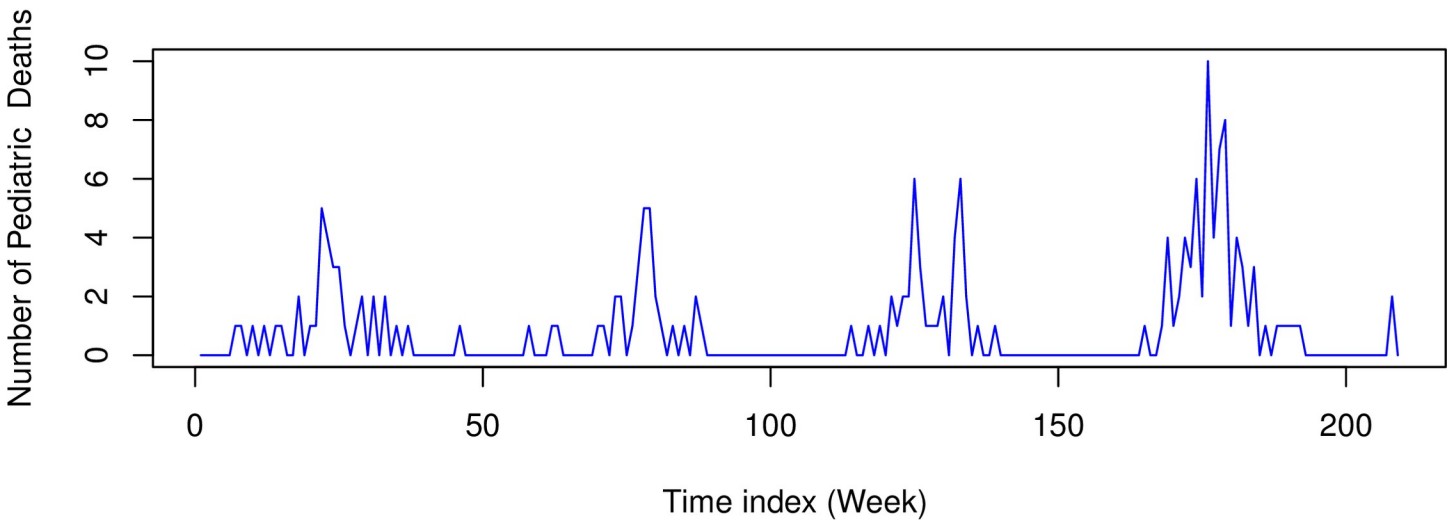

## Weekly Average of Nationwide Low Temperature

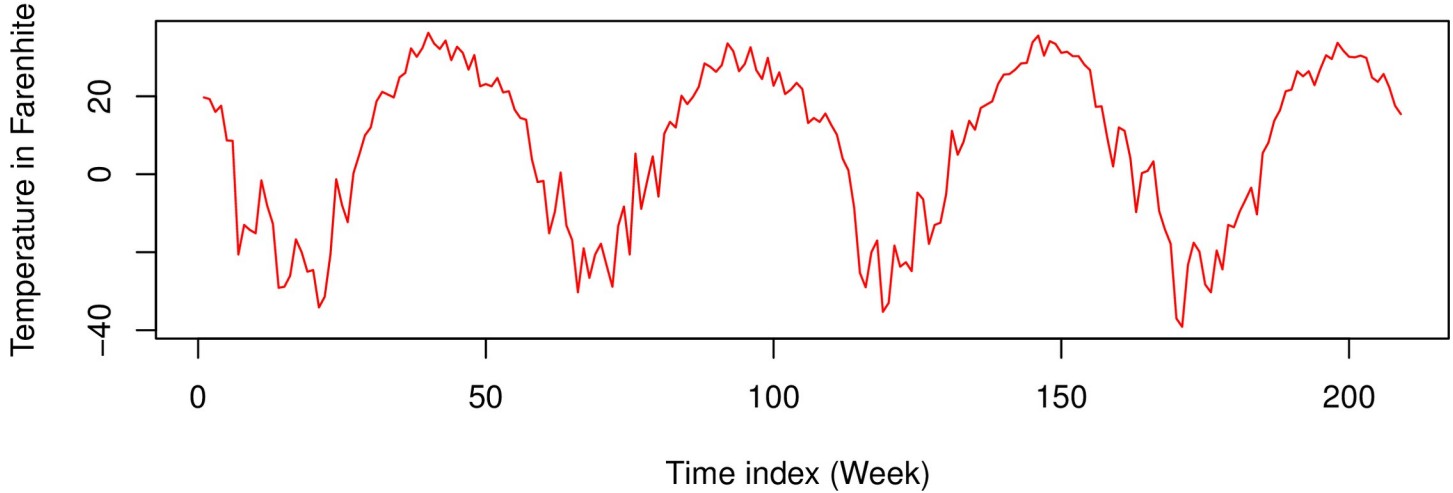

**Fig 7. Influenza B associated pediatric deaths and weekly average of nationwide low temperature.** (Blue) Time series plots of Influenza B associated pediatric deaths. (Red) Weekly average of nationwide low temperature.

inflated component in the pediatric mortality data set. In this example, time-varying zero-inflation process was modelled using the formulation from Eq (6) with low temperature as the input variable to the logistic model. We label this zero-inflation scenario as Sc4. Results from this the TVZIP-GARCH formulation under model orders M1, M2, and M3, with varying underlying distributions are given in Table 15. Note that we used the EM algorithm as was done in Example 1.

Results in Table 15 show that the TVZIP-INARCH (2) model fitted under Scenario 4 (Sc4) exhibited the lowest AIC and BIC values compared to the other models.

**Table 15.** Estimated parameters, standard errors (within parentheses), AIC and BIC for the pediatric death counts cause by virus B.

| Model | Estimated parameters and standard errors | | | | | | Model selection criteria | |
|---|---|---|---|---|---|---|---|---|
| | $\hat{\delta}_0$ | $\hat{\delta}_1$ | $\hat{\alpha}_0$ | $\hat{\alpha}_1$ | $\hat{\alpha}_2$ | $\hat{\beta}_1$ | AIC | BIC |
| Sc4 C0 M1 | -1.7436 (0.3280) | 0.1115 (0.0158) | 0.7258 (0.1040) | 0.5789 (0.0834) | | | 453.1119 | 466.4812 |
| **Sc4 C0 M2** | -2.5485 (0.4310) | 0.1190 (0.0189) | 0.3478 (0.0800) | 0.4027 (0.0812) | 0.4101 (0.0831) | | **427.7886** | **444.5003** |
| Sc4 C0 M3 | -1.0832 (0.2539) | 0.1005 (0.0131) | 0.1797 (0.1909) | 0.4090 (0.0947) | | 0.2493 (0.0680) | 441.1931 | 457.9048 |

Forty five models were fitted under the constant zero-inflation scenario (Sc3). Table 1B in S6 Appendix provides list of these models, which fall into categories C1, C2, C4, and C6 with model orders M1, M2, and M3. The zero-inflated versions of Poisson, negative binomial Type 1, negative binomial Type 2, and generalized Poisson distributions we considered as the underlying distributions associated witn the counts in categories C1, C4, and C6. In C2, we assumed the zero-inflated geometric Poisson and zero-inflated Neyman Type A as the underlying distributions. As was done in Example 1, a test for a change point was conducted if a model had the lowest AIC value under each model category C1 by model order combination. Note that at this initial stage, models with the lowest AIC value under the C1 category had the lowest BIC value as well. If such a point was detected, the model under consideration was independently fitted to sub-series before and after the change point. Comparison of the versions of the proposed model with the forty-five existing models, as well as for models fitted to sub-series, are repoted in Table 1A and 1B in S6 Appendix. The models with the lowest AIC or BIC values in each model category by model order combination are highlighted in light gray. Models with the lowest AIC or BIC values within each model category are identifies in boldface type in Tables 1A and 1B in S6 Appendix. If the model has the lowest information criteria values with respoect to both AIC and BIC, then it is identified in italic boldface font.

The results for TVZIP-INGARCH ($p$, $q$) models under Scenario 4 (Sc4) together with the contant zero-inflation models under Scenario 3 (Sc3) that had the lowest AIC or BIC values within each model category are shown in Table 16.

Based on the BIC criterion, TVZIP-INARCH (2) model provided the best fit to the data, albeit by a marginally lower BIC value compared to the next best model. However, a different result is obtained when considering the AIC values. AIC values indicate that the ZINB2 softplus INARCHX (2) model, which assumes a constant zero-inflation, exhibits a marginally better fit to the data. This is the new model we introduced as a generalization of the softplus INGARCH formulation of Weiß et al. [12] by including zero-inflation and an exogenous variable to the original version. The inclusion of an exogenous variable seems to lead to an enhanced performance with respect to AIC. This advantage vanishes slightly when BIC is used as a model selection criterion. Since both models perform well, model adequacy checks were

**Table 16.** The AIC and BIC values for the best model in model category C1, C2, C4 and C6 with scenario Sc3 and Sc4.

| Zero-inflated scenario | Model information | | Model selection criteria | |
|---|---|---|---|---|
| | Model category | Model order | AIC | BIC |
| Sc4 | C0: Time-varying zero-inflated Poisson INGARCH | **TVZIP-INARCH (2)** | **427.7886** | **444.4522** |
| Sc3 | C1: Zero-inflated INGARCH | ZINB2-INARCH (2) | 430.6082 | 447.2718 |
| | | ZINB2-INARCH (2)—change point | 428.0244 | 454.1387 |
| | C2: Zero-inflated compound Poisson INGARCH | ZIGEOMP-INARCH (2) | 433.8181 | 450.4817 |
| | C4: Zero-inflated log-linear INGARCHX | ZINB2 log-linear INARCHX (2) | 429.7920 | 449.7883 |
| | C6: Zero-inflated softplus INGARCHX | **ZINB2 softplus INARCHX (2)** | **425.5864** | **445.5827** |

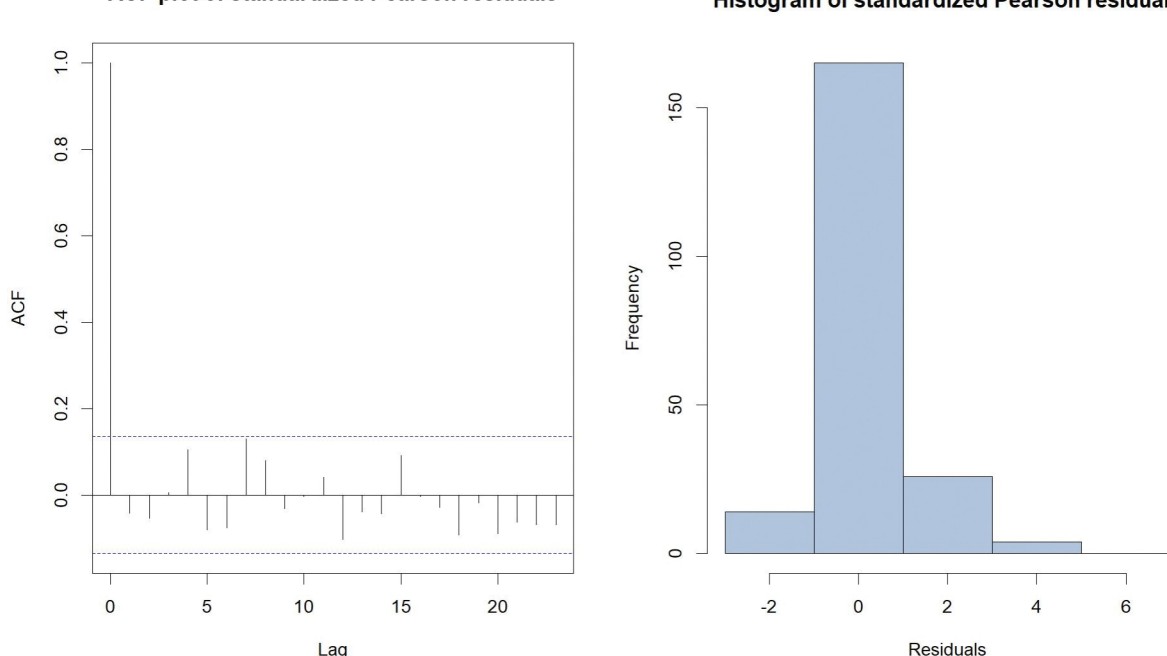

**Fig 8. The ACF plot and the histogram, both based on the standardized Pearson residuals of fitted TVZIP-INARCH (2) model.**

carried out for the TVZIP-INARCH (2) and the ZINB2 softplus INARCHX (2) models. The ACF plots, and the histograms of standardized Pearson residuals for the fitted models are presented in Figs 8 and 9 respectively.

The ACF plots indicate that the standardized Pearson residuals of both TVZIP-INARCH (2) and the ZINB2 softplus INARCHX (2) models do not exhibit significant autocorrelations. However, the mean (-0.0054) and the variance (1.0006) of the residuals of the TVZIP-INARCH (2) model are closer to 0 and 1 respectively. The residuals of the ZINB2 softplus INARCHX (2) model have a mean (-0.0389) close to 0, but variance (0.8557) is further away from 1 compared to the variance of the TVZIP-INARCH (2) Model. While both models satisfied the residual checks for model adequacy, the TVZIP-INARCH (2) may be considered to perform marginally better due to its slightly lower BIC value and the residual mean and variance being closer to the ideal values of 0 and 1 respectively. In addition to that, the mean of the fitted values of TVZIP-INARCH (2) model (0.8421) is closer to the sample mean (0.8517) than that of the ZINB2 softplus INARCHX (2) model (0.9264). Thus, we can conclude that while both these models performed well, our proposed TVZIP-INGARCH model has a tenuous edge over the Softplus INGARCHX model. In a practical sense, however, one can argue that both models perform equally well.

One reason for the good performance of ZINB2 softplus INARCHX model may be due to the nature of the softplus function that can bring the value of the conditional mean close to zero for some values of the exogenous variable. Thus, while the zero-inflated component was set up to be non-time-varying, the influence of the exogenous variable can bring the conditional mean of the process very close to zero over some periods. In other words, softplus link function allows the exogenous variable to significantly increase the probability of obtaining zero counts in a time-varying fashion.

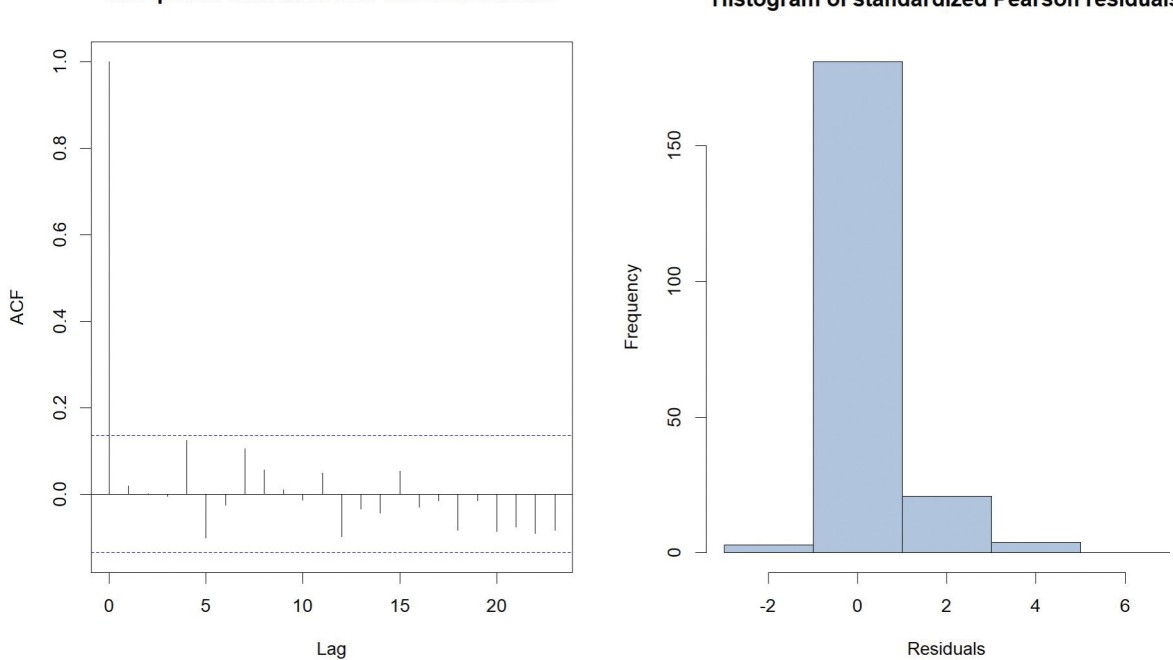

**Fig 9. The ACF plot and the histogram, both based on the standardized Pearson residuals of fitted ZINB2 softplus INARCHX (2) model.**

The main aim of this example is to illustrate the utility of the proposed formulation in modelling a real-life situation. Based on the results from this example, TVZIP-INARCH (2) formulation with zero-inflation modeled using an exogenous variable and the ZINB2 softplus INARCHX (2) model appear as good candidates to be included in the tool set a practitioner may consider when modeling count data time series with time-varying zero inflation, in situations where an appropriate exogenous variable is available. Equally important is the observation that no other model outperformed the proposed model with respect to any criteria based on residual analysis.

## Conclusions

A time-varying zero-inflated Poisson integer GARCH (TVZIP-INGARCH) model was proposed to accommodate situations where zero-inflation is driven by either a deterministic function of time or a set of exogenous variables. Monte-Carlo simulation results indicate that the Expected Maximization (EM) and Maximum Likelihood Estimation (MLE) methods produce very similar results with respect to parameter estimates. It is observed that both EM and MLE techniques estimated the model parameters with good accuracy when the underlying model has a purely ARCH component. When the model has a GARCH component, the GARCH parameters are estimated with lesser accuracy, which is a phenomenon also seen in the study of the existing ZIP-INGARCH model proposed by Zhu [2]. When tested on two real-life data sets, the TVZIP-INGARCH models performed better than the other constant zero-inflated INGARCH formulations with one possible exception. Even in this case the competing model does not outperform the proposed model. These results illustrate the usefulness of the proposed model as one of the tools the practitioner can utilize for modeling empirical count data time series with time-varying zero-inflation. In addition, the flexibility of modelling zero-

inflation through deterministic cyclical functions or through exogenous time series provide the proposed model added versatility.

## Supporting information

**S1 Appendix. Derivation of conditions for zero-inflation probability to lie inside (0,1).**
(DOCX)

**S2 Appendix. Derivation of the conditional mass function.**
(DOCX)

**S3 Appendix. Initialization of the lambdas.**
(DOCX)

**S4 Appendix. List of existing INGARCH type models used in the comparison study.**
(DOCX)

**S5 Appendix. Model comparison results for the pediatric death counts cause by virus A.**
(DOCX)

**S6 Appendix. Model comparison results for the pediatric death counts cause by virus B.**
(DOCX)

**S1 Data. Datasets.**
(XLSX)

**S1 File. Codes.** This file contains the codes for simulation study, real data example 1, and real data example 2.
(ZIP)

## Acknowledgments

The authors wish to thank the academic editor and the reviewers for their insightful and constructive comments, which helped to greatly enhance the quality and breath of this paper.

## Author Contributions

**Conceptualization:** V. A. Samaranayake.

**Formal analysis:** Isuru Panduka Ratnayake.

**Methodology:** Isuru Panduka Ratnayake, V. A. Samaranayake.

**Software:** Isuru Panduka Ratnayake.

**Supervision:** V. A. Samaranayake.

**Validation:** Isuru Panduka Ratnayake.

**Visualization:** Isuru Panduka Ratnayake.

**Writing – original draft:** Isuru Panduka Ratnayake.

**Writing – review & editing:** Isuru Panduka Ratnayake, V. A. Samaranayake.

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
