## [Decision Letter · Decision Letter 0]

1 Nov 2022

PONE-D-22-26550An integer GARCH model for a Poisson process with time-varying zero-inflationPLOS ONE

Dear Dr. Ratnayake,

Thank you for submitting your manuscript to PLOS ONE. After careful consideration, we feel that it has merit but does not fully meet PLOS ONE’s publication criteria as it currently stands. Therefore, we invite you to submit a revised version of the manuscript that addresses the points raised during the review process.

Please write a concise abstract, i.e., shorten your abstract. The abstract should be self-contained and entirely understandable without reference to other sources.For the real examples, more models should be compared, i.e., the proposed models should be compared with other zero-inflated INGARCH models proposed in the literature.For your final model choice, please provide appropriate adequacy checks. Please provide the ACF plot and histogram of the standardized Pearson residuals, i.e., if their mean is close to 0, their variance close to 1, and they are serially uncorrelated.Please ensure that your decision is justified on PLOS ONE’s publication criteria and not, for example, on novelty or perceived impact.

We look forward to receiving your revised manuscript.

Kind regards,

Cathy W. S. Chen, Ph.D.

Academic Editor

PLOS ONE

Journal Requirements:

Additional Editor Comments

1. Please write a concise abstract, i.e., shorten your abstract. The abstract should be self-contained and entirely understandable without reference to other sources.

2. For the real examples, more models should be compared, i.e., the proposed models should be compared with other zero-inflated INGARCH models proposed in the literature.

3. For your final model choice, please provide appropriate adequacy checks. Please provide the ACF plot and histogram of the standardized Pearson residuals, i.e., if their mean is close to 0, their variance close to 1, and they are serially uncorrelated.

Reviewers' comments:

Reviewer's Responses to Questions

**Comments to the Author**

1. Is the manuscript technically sound, and do the data support the conclusions?

Reviewer #1: Yes

Reviewer #2: Partly

2. Has the statistical analysis been performed appropriately and rigorously? 

Reviewer #1: Yes

Reviewer #2: Yes

3. Have the authors made all data underlying the findings in their manuscript fully available?

Reviewer #1: Yes

Reviewer #2: Yes

4. Is the manuscript presented in an intelligible fashion and written in standard English?

Reviewer #1: Yes

Reviewer #2: Yes

5. Review Comments to the Author

Reviewer #1: The topic is interesting and the paper is well written. Some comments are given as follows.

1. The idea (i.e., time-varying zero-inflation) in this paper can be dated back to Lambert (1992), which should be mentioned.

2. The references are not new, the newest one is in 2012, except Xu et al. (2020), so the Introduction part should be rewritten and pay more attention to recent references. For example, Xiong and Zhu (2019) and Li et al. (2021) considered robust estimation methods for INGARCH models, Weiss et al. (2022) proposed the softplus link function as an alternative to identity and log link functions when constructing the INGARCH models, Liu et al. (2022) generalized the range of observations from infinite to categorical, Cui et al. (2021) and Xu and Zhu (2020) generalized the INGARCH models from count-valued to the Z-valued cases, just mention some among others. In addition, you can mention a recent review paper: Davis et al. (2021).

Some early references can be deleted to save spaces.

3. For the real examples, more models should be compared, i.e., the proposed models should be compared with other zero-inflated INGARCH models proposed in the literature, such as those in Goncalves et al. (2016), Xu et al. (2020) and Lee et al. (2021).

4. Using temperature as a covariate is not new in modeling time series of counts, such as Zhu and Wang (2015) and Chen and Lee (2017), where they used the log-linear INGARCH models. In fact, the softplus INGARCH model in Weiss et al. (2022) can also include a covariate. I wonder, the proposed models and the common zero-inflated versions of the above log-linear and softplus models (i.e., with a constant zero-inflated probability), which is better when analyzing these real datasets?

5. Minor comments. p.3, “Andreas Heinen [3]” should be “Heinen [3]”; p.4, “Integer GARCH” should be “Integer-valued GARCH”; p.4, “related the proposed” should be “related to the proposed”; p.8, “time depended” should be “time-dependent”; p.34, “is available” should be “are available”; p.35, “zero-inflated poisson” should be “zero-inflated Poisson”; p.36, “there deterministic”?

[1] Chen, C.W. S. and Lee, S. (2017). Bayesian causality test for integer-valued time series models with applications to climate and crime data. Journal of the Royal Statistical Society Series C, 66, 797–814.

[2] Cui, Y., Li, Q. and Zhu, F. (2021). Modeling Z-valued time series based on new versions of the Skellam INGARCH model. Brazilian Journal of Probability and Statistics, 35, 293-314.

[3] Davis, R.A., Fokianos, K., Holan, S.H., Joe, H., Livsey, J., Lund, R., Pipiras, V. and Ravishanker, N. (2021). Count time series: A methodological review. Journal of the American Statistical Association, 116, 1533-1547.

[4] Goncalves, E., Mendes-Lopes, N., Silva, F. (2016). Zero-inflated compound Poisson distributions in integer-valued GARCH models. Statistics, 50, 558–578.

[5] Lambert, D. (1992). Zero-inflated Poisson regression, with an application to defects in manufacturing. Technometrics, 34, 1-14.

[6] Lee, S., Kim, D., Seok, S. (2021). Modeling and inference for counts time series based on zero-inflated exponential family INGARCH models. Journal of Statistical Computation and Simulation, 91, 2227–2248.

[7] Li, Q., Chen, H. and Zhu, F. (2021). Robust estimation for Poisson integer-valued GARCH models using a new hybrid loss. Journal of Systems Science and Complexity, 34, 1578-1596.

[8] Liu, M., Zhu, F. and Zhu, K. (2022). Modeling normalcy-dominant ordinal time series: An application to air quality level. Journal of Time Series Analysis, 43, 460-478.

[9] Weiss, C.H., Zhu, F. and Hoshiyar, A. (2022). Softplus INGARCH models. Statistica Sinica, 32, 1099-1120.

[10] Xiong, L. and Zhu, F. (2019). Robust quasi-likelihood estimation for the negative binomial integer-valued GARCH(1,1) model with an application to transaction counts. Journal of Statistical Planning and Inference, 203, 178-198.

[11] Xu, Y. and Zhu, F. (2022). A new GJR-GARCH model for Z-valued time series. Journal of Time Series Analysis, 43, 490-500.

[12] Zhu, F. and Wang, D. (2015). Empirical likelihood for linear and log-linear INGARCH models. Journal of the Korean Statistical Society, 44, 150-160.

Reviewer #2: I have read the manuscript entitled "An integer GARCH model for a Poisson process with time-varying zero-inflation", which has been submitted for possible publication in PLOS ONE. The article is generally well written and considers a relevant topic, but it also exhibits a couple of deficiencies that need to be solved prior to publication. Thus, the authors should prepare a comprehensive revision, where all my comments are carefully addressed.

1) p. 6: When discussing some non-linear versions of INGARCH models, also the recent softplus INGARCH model is worth mentioning, which behaves nearly like a linear INGARCH model, but allows for negative autocorrelations, see Weiß et al. (2022): Softplus INGARCH Models.

Statistica Sinica 32(2), 1099-1120.

2) p. 7: It seems that the NB-INGARCH model proposed by Ye et al. [22] coincides with the one proposed by

Xu, H.-Y., Xie, M., Goh, T.N., Fu, X. (2012) A model for integer-valued time series with conditional overdispersion. Computational Statistics and Data Analysis 56(12), 4229–4242.

but differs from the one of

Zhu, F. (2011) A negative binomial integer-valued GARCH model. Journal of Time Series Analysis 32(1), 54–67.

3) It takes until p. 8 (l. 168) until you describe your contribution, i.e., the literature review is extremely long. I suggest to shorten it at least with respect to non-INGARCH models, such as INAR or GLARMA. Maybe also some rather special INARCH models (e.g., with a double-Poisson conditional distribution) or the topic of interventions could be omitted without loss. For such alternative approaches, you can just refer to some introductory textbook on discrete-valued time series.

4) p. 10, l. 202: The Statement "The dynamic propagation of the conditional mean of the Poisson process is defined by" is not correct, because the conditional mean is not lambda_t, but lambda_t*(1-omega_t). In fact, if you want the conditional to be influenced only by past counts but not directly by the exogeneous information, you would need to reparametrize the ZIP distribution by mu and omega, where mu=lambda(1-omega).

5) p. 10, l. 217, "See S1 Appendix, for the derivation of the conditional mean and conditional variance.":

This derivation is really not necessary, because mean and variance of the ZIP distribution (and thus conditional mean and variance of your model) are well known. You can find it in any textbook on count data or discrete-valued time series, or also in the famous book by Johnson and Kotz on univariate discrete distribution. So just provde an appropriate reference here.

6) The last inequality in formula (4) is wrong, it is an equality.

7) p. 17, l. 323, statement "With a reasonable

initial starting value": Please be more precise here. If q>0, you need to specify the first few values of lambda, and it is well-known that the resulting estimates are extremely sensitive to this choice, see the discussion on p. 78 in Weiß (2018), An Introduction to Discrete-valued Time Series, Wiley.

8) Regarding your Simulation Study, on the one hand, one would expect that EM and ML end up with (roughly) the same estimates, at least after a sufficient number of iterations. But maybe also because of sich initialization effects, this is not always the case. Besides the actual estimation performance, another criterion for evaluating the methods would be their computational effort. Can you provide and discuss computing times for your simulations?

By the way, when you observe "that the mean of the estimates are not very close to the true values even with higher sample sizes" (p. 26), this might have been caused by an inappropriate initialization of the lambdas.

9) For your data examples, you do model selection by information criteria, which is OK, but which does not give any insights into model adequacy. Therefore, for your final model choice, please provide appropriate adequacy checks. I would at least expect an analysis of the standardized Pearson residuals, i.e., if their mean is close to 0, their variance close to 1, and that they are serially uncorrelated. If there are deviations, these should be explained, and possible (future) solution for solving the issues should be sketched.

6. PLOS authors have the option to publish the peer review history of their article (what does this mean?). If published, this will include your full peer review and any attached files.

Reviewer #1: No

Reviewer #2: No

---

## [Author Response · Author response to Decision Letter 0]

30 Mar 2023

Thank you for the invitation to submit a revised version of our manuscript (PONE-D-22-26550). We wish to thank the editor and the reviewers for the insightful comments.

Editor comments

We considered all points raised by the you and reviewers, and carefully addressed them as described below. 

1. We shortened the abstract and the new version of the abstract is entirely understandable without reference to other sources. 

2. In the real example comparisons, the proposed model is compared with other zero-inflated models available in the literature as suggested by the reviewers. In addition, based on a comment of a reviewer we implemented constant zero-inflation versions of an existing log-linear INGARCH with and without exogenous variable inputs. Moreover, we generalized the softplus INGARCH model of Weiß et al. (2022), by introducing zero inflation and an exogenous variable. These existing and modified models were run under the assumption of zero-inflated Poisson, zero-inflated negative binomial (Type1 and Type 2 as classified by Zhu’s 2012 paper) and zero-inflated generalized Poisson. In total we compared our proposed model with 45 zero-inflated INGARCH type models in each separate example, some of which are extended versions of existing models as described above. Across both real-life examples, a total of 69 unique models were used in addition to variations of our proposed model. 

3. For the final model choice, we provided appropriate adequacy checks as suggested by you and a reviewer. This includes ACF plot and histogram of the standardized Pearson residuals.

Reviewer 1 comments 

Thank you very much for your detailed and useful comments. We have addressed each of them as follows.

1. The idea (i.e., time-varying zero-inflation) in this paper can be dated back to Lambert (1992), which should be mentioned.

Response: We thank Reviewer 1 for this valuable suggestion, and we agree. We have included this to the Introduction (p. 6 lines 124-125). 

2. The references are not new, the newest one is in 2012, except Xu et al. (2020), so the Introduction part should be rewritten and pay more attention to recent references. For example, Xiong and Zhu (2019) and Li et al. (2021) considered robust estimation methods for INGARCH models, Weiß et al. (2022) proposed the softplus link function as an alternative to identity and log link functions when constructing the INGARCH models, Liu et al. (2022) generalized the range of observations from infinite to categorical, Cui et al. (2021) and Xu and Zhu (2020) generalized the INGARCH models from count-valued to the ℤ-valued cases, just mention some among others. In addition, you can mention a recent review paper: Davis et al. (2021). Some early references can be deleted to save spaces. 

Response: We thank Reviewer 1 for this valuable suggestion. We regret that our literature review was somewhat outdated. Accordingly, we have added relevant studies in the Introduction (p.7, lines 138-147). Now we believe that more newest and relevant references are included in the literature review. 

3. For the real examples, more models should be compared, i.e., the proposed models should be compared with other zero-inflated INGARCH models proposed in the literature, such as those in Goncalves et al. (2016), Xu et al. (2020) and Lee et al. (2021). 

Response: We thank reviewer 1 for this suggestion and we agree. We believe this helped us to improve the quality of our manuscript. Accordingly, the proposed model’s performance is compared with constant zero-inflated probability versions of the INGARCH (p, q) model such as zero-inflated Poisson (ZIP), zero-inflated negative binomial Type 1(ZINB1), zero-inflated negative binomial Type 2 (ZINB2), all proposed by Zhu (2012), and the zero-inflated generalized Poisson (ZIGP) model proposed by Chen and Lee (2016). The zero-inflated compound Poisson INGARCH models such as zero-inflated geometric Poisson INGARCH (ZIGEOMP-INGARCH) and the zero-inflated Neyman Type A INGARCH (ZINTA-INGARCH) of Gonçalves et al. (2016), were also fitted to the data. In addition, the following formulations were also included in the comparison. The log-linear INGARCH model of Fokianos and Tjøstheim (2011) was modified to include zero-inflation. We also modified the log-linear INGARCHX model of Chen and Lee (2017) to include zero-inflation, but it should be noted that with the inclusion of zero-inflation the above two models are nested within the non-time-varying version of the model of Xu et al. (2020). We extended the softplus INGARCH model introduced by Weiß et al. (2022), by introducing zero inflation and an exogenous variable. The INGARCH, log-linear INGARCH, log-linear INGARCHX, the softplus INGARCH, and the softplus INGARCHX models were fitted assuming the zero-inflated versions of the Poisson, negative binomial Type 1, negative binomial Type 2, and generalized Poisson distributions. For definitions of negative binomial Type 1 and Type 2 distributions. Note that the compound Poisson model of Gonçalves et al. (2016) was fitted assuming the zero-inflated geometric Poisson and the zero-inflated Neyman Type A distributions. The above combinations of model categories and distributions yield a total of twenty-three count data time series formulations. These twenty-three formulations are listed in Table 1 in S4 appendix.

4. Using temperature as a covariate is not new in modeling time series of counts, such as Zhu and Wang (2015) and Chen and Lee (2017), where they used the log-linear INGARCH models. In fact, the softplus INGARCH model in Weiss et al. (2022) can also include a covariate. I wonder, the proposed models and the common zero-inflated versions of the above log-linear and softplus models (i.e., with a constant zero-inflated probability), which is better when analyzing these real datasets?

Response: We are grateful for this suggestion, and we agree. We also fitted the constant zero-inflated versions of log-linear INGARCH (p, q) and softplus INGARCH (p, q) models in Real data example - Use of a deterministic sinusoidal zero-inflation function and Real data example. More results are given in Table 1B in S5 Appendix. In this example the proposed model outperforms these non-linear INGARCH models with respect to all criteria. Moreover, as suggested by the reviewer we also fitted zero-inflated versions of log-linear INGARCHX (p, q) and zero-inflated versions of softplus INGARCHX (p, q) models to Real data example - Zero-inflation function is driven by exogenous variable. In this example, temperature was introduced as the exogenous variable to the INGARCH formulations. More details can be found in Table 1B in S6 Appendix. The results indicate that the extended version of the zero-inflated softplus INGARCHX model performed as well as our proposed model with respect to some criteria, but not with respect to all. 

5. Minor comments. p.3, “Andreas Heinen [3]” should be “Heinen [3]”; p.4, “Integer GARCH” should be “Integer-valued GARCH”; p.4, “related the proposed” should be “related to the proposed”; p.8, “time depended” should be “time-dependent”; p.34, “is available” should be “are available”; p.35, “zero-inflated poisson” should be “zero-inflated Poisson”; p.36, “there deterministic”?

Response: We are grateful for reviewer 1 for these recommendations and we are sorry to make these mistakes. We addressed these comments as recommended. “Andreas Heinen [3]” was changed to “Heinen [5]” (p.4, line 66); “Integer GARCH” was corrected to “Integer-valued GARCH” (p.4, line 74); “time depended” was corrected as “time-dependent” (p.8 line 161); “is available” changed to “are available” (p.41 line 638). The typos such as “zero-inflated poisson” was replaced to “zero-inflated Poisson” everywhere it appears. Since some sections were removed from the previous manuscript, typos like “there deterministic” were omitted. 

Reviewer 2 Comments 

We appreciate the detailed feedback that you provided, which helped us improve the quality of our submission notably.

1.p. 6: When discussing some non-linear versions of INGARCH models, also the recent softplus INGARCH model is worth mentioning, which behaves nearly like a linear INGARCH model, but allows for negative autocorrelations, see Weiß et al. (2022): Softplus INGARCH Models. Statistica Sinica 32(2), 1099-1120.

Response: We appreciate reviewer 2 for this comment and we agree. We discussed softplus INGARCH model in the literature review (p 5-p6 lines 107-108). Additionally, we extended the softplus INGARCH model introduced by Weiß et al. (2022), by introducing zero inflation and an exogenous variable. Moreover, we employed these modified zero-inflated versions of softplus INGARCH and softplus INGARCHX models in the real-world examples. We also compared proposed model’s performance with zero-inflated softplus INGARCH model (in Real data example - Use of a deterministic sinusoidal zero-inflation function and Real data example) and zero-inflated softplus INGARCHX model (Real data example - Use of a deterministic sinusoidal zero-inflation function and Real data example - Zero-inflation function is driven by exogenous variable). 

2.p. 7: It seems that the NB-INGARCH model proposed by Ye et al. [22] coincides with the one proposed by Xu, H.-Y., Xie, M., Goh, T.N., Fu, X. (2012) A model for integer-valued time series with conditional overdispersion. Computational Statistics and Data Analysis 56(12), 4229–4242. but differs from the one of Zhu, F. (2011) A negative binomial integer-valued GARCH model. Journal of Time Series Analysis 32(1), 54–67.

Response: We are grateful for this comment. Due to this input, we found a mistake we did in our initial literature review and corrected it. We included this section to the introduction section (p. 6, lines 114-123) in this version of the manuscript. 

3.It takes until p. 8 (l. 168) until you describe your contribution, i.e., the literature review is extremely long. I suggest to shorten it at least with respect to non-INGARCH models, such as INAR or GLARMA. Maybe also some rather special INARCH models (e.g., with a double-Poisson conditional distribution) or the topic of interventions could be omitted without loss. For such alternative approaches, you can just refer to some introductory textbook on discrete-valued time series 

Response: We thank Reviewer 2 for this valuable suggestion. We regret that our literature review was extremely long. We modified our literature review as recommended and We introduced our contribution in page 3 (lines 44-61). Moreover, we removed previously discussed non-INGARCH models (e.g., INAR and GLARMA) and some special INGARCH models (e.g., with a double-Poisson conditional distribution) from the Introduction. 

4. p. 10, l. 202: The Statement "The dynamic propagation of the conditional mean of the Poisson process is defined by" is not correct, because the conditional mean is not lambda_t, but lambda_t*(1-omega_t). In fact, if you want the conditional to be influenced only by past counts but not directly by the exogeneous information, you would need to reparametrize the ZIP distribution by mu and omega, where mu=lambda(1-omega).

Response: We are grateful reviewer 2 for this comment. We agreed with reviewer 2. We reparametrized the ZIP distribution with μ_tand ω_t(i.e., ZIP(μ_t,ω_t )). Now the conditional mean of the Poisson process is defined by μ_t=(1-ω_t ) λ_t, where is the intensity parameter of the Poisson process. These changes are included in the manuscript (p. 10 lines 208 - 216) 

5. p. 10, l. 217, "See S1 Appendix, for the derivation of the conditional mean and conditional variance.": This derivation is really not necessary, because mean and variance of the ZIP distribution (and thus conditional mean and variance of your model) are well known. You can find it in any textbook on count data or discrete-valued time series, or also in the famous book by Johnson and Kotz on univariate discrete distribution. So just provide an appropriate reference here.

Response: We thank reviewer 2 for this suggestion and we agree. We removed the S1 Appendix. 

6. The last inequality in formula (4) is wrong, it is an equality. 

Response: We are grateful reviewer 2 for pointing out this mistake and we agree. We regret that we did this mistake and we corrected it. (p.11 line 232)

7. p. 17, l. 323, statement "With a reasonable initial starting value": Please be more precise here. If q>0, you need to specify the first few values of lambda, and it is well-known that the resulting estimates are extremely sensitive to this choice, see the discussion on p. 78 in Weiß (2018), An Introduction to Discrete-valued Time Series, Wiley.

Response: We are thankful reviewer 2 for this valuable suggestion, which we agreed and we believed this improved the quality of our submission. We addressed this through by adding a paragraph to the manuscript (p. 18, lines 341-353). In addition to that empirical evidences are included in S3 Appendix.

8. Regarding your Simulation Study, on the one hand, one would expect that EM and ML end up with (roughly) the same estimates, at least after a sufficient number of iterations. But maybe also because of such initialization effects, this is not always the case. Besides the actual estimation performance, another criterion for evaluating the methods would be their computational effort. Can you provide and discuss computing times for your simulations? By the way, when you observe "that the mean of the estimates are not very close to the true values even with higher sample sizes" (p. 26), this might have been caused by an inappropriate initialization of the lambdas.

Response: We are thankful reviewer 2 for this input and we agreed. We calculated the computational effort for each simulation run and presented these results in Table 1 -Table 12. We also discussed the initialization of the lambdas in S3 Appendix. In addition to that we mentioned this in p.28 lines 462-464 and p. 36 lines 521-522.

9. For your data examples, you do model selection by information criteria, which is OK, but which does not give any insights into model adequacy. Therefore, for your final model choice, please provide appropriate adequacy checks. I would at least expect an analysis of the standardized Pearson residuals, i.e., if their mean is close to 0, their variance close to 1, and that they are serially uncorrelated. If there are deviations, these should be explained, and possible (future) solution for solving the issues should be sketched.

Response: We are grateful reviewer 2 for suggesting model adequacy checks. We agreed and believed that this improved our manuscript. As suggested for the final model choice we calculated the standardized Pearson residuals, analyzed the mean, and variance. In addition to that the histogram and the ACF plot were constructed (see Fig 4, Fig 8, and Fig 9) and the interpretations were provided. 

Reference

1.Cathy W.S. Chen, Sangyeol Lee, Generalized Poisson autoregressive models for time series of counts, Computational Statistics & Data Analysis, Volume 99,2016, Pages 51-67,ISSN 0167-9473,https://doi.org/10.1016/j.csda.2016.01.009

2. Chen, C.W. S. and Lee, S. (2017). Bayesian causality test for integer-valued time series models with applications to climate and crime data. Journal of the Royal Statistical Society Series C, 66, 797–814.

3.Cui, Y., Li, Q. and Zhu, F. (2021). Modeling Z-valued time series based on new versions of the Skellam INGARCH model. Brazilian Journal of Probability and Statistics, 35, 293-314.

4.Davis, R.A., Fokianos, K., Holan, S.H., Joe, H., Livsey, J., Lund, R., Pipiras, V. and Ravishanker, N. (2021). Count time series: A methodological review. Journal of the American Statistical Association, 116, 1533-1547.

5.Esmeralda Gonçalves, Nazaré Mendes-Lopes & Filipa Silva. Zeroinflated compound Poisson distributions in integer-valued GARCH models, Statistics, 2015 DOI: 10.1080/02331888.2015.1114622

6.Fokianos K, Tjøstheim D. Log-linear Poisson autoregression. J Multivar Anal. 2011;102(3):563-78.

7.Lambert, D. (1992). Zero-inflated Poisson regression, with an application to defects in manufacturing. Technometrics, 34, 1-14.

8.Li, Q., Chen, H. and Zhu, F. (2021). Robust estimation for Poisson integer-valued GARCH models using a new hybrid loss. Journal of Systems Science and Complexity, 34, 1578-1596.

9. Lee, S., Kim, D., Seok, S. (2021). Modeling and inference for counts time series based on zero-inflated exponential family INGARCH models. Journal of Statistical Computation and Simulation, 91, 2227–2248. 

10.Liu, M., Zhu, F. and Zhu, K. (2022). Modeling normalcy-dominant ordinal time series: An application to air quality level. Journal of Time Series Analysis, 43, 460-478.

11. Weiß, C.H. An Introduction to Discrete-Valued Time Series; John Wiley & Sons: Hoboken, NJ, USA, 2018.

12. Weiß, C.H.; Zhu, F.; Hoshiyar, A. Softplus INGARCH models. Stat. Sin. 2022, 32, 1099–1120.

13. Xiong, L. and Zhu, F. (2019). Robust quasi-likelihood estimation for the negative binomial integer-valued GARCH (1,1) model with an application to transaction counts. Journal of Statistical Planning and Inference, 203, 178-198.

14.Xu, H.-Y.; Xie, M.; Goh, T.N.; Fu, X. A Model for Integer–Valued Time Series With Conditional Overdispersion. Comput. Stat. Data Anal. 2012, 56, 4229–4242.

15. Xu X, Chen Y, Chen CWS, Lin X. Adaptive log-linear zero-inflated generalized poisson autoregressive model with applications to crime counts. Ann Appl Stat. 2020;14(3):1493-515.

16. Ye F, Garcia TP, Pourahmadi M, Lord D. Extension of negative binomial GARCH model analyzing effects of gasoline price and miles traveled on fatal crashes involving intoxicated drivers in Texas. Transp Res Rec. 2012;2279(1):31-9.

17. Y. Xu, F. Zhu, A new GJR-GARCH model for Z-valued time series, J. Time Ser. Anal., 2022, in press. http://dx.doi.org/10.1111/jtsa.12623

18. Zhu F. A negative binomial integer-valued GARCH model. J Time Ser Anal. 2011;32(1):54-67.

19. Zhu F. and Wang, D. (2015). Empirical likelihood for linear and log-linear INGARCH models. Journal of the Korean Statistical Society, 44, 150-160. 

20. Zhu F. Zero-inflated Poisson and negative binomial integer-valued GARCH models. J Stat Plan Inference. 2012a;142(4):826-39.

---

## [Decision Letter · Decision Letter 1]

10 Apr 2023

PONE-D-22-26550R1An integer GARCH model for a Poisson process with time-varying zero-inflationPLOS ONE

Dear Dr. Ratnayake,

Thank you for submitting your manuscript to PLOS ONE. After careful consideration, we feel that it has merit but does not fully meet PLOS ONE’s publication criteria as it currently stands. Therefore, we invite you to submit a revised version of the manuscript that addresses the points raised during the review process.

The abstract should be self-contained and entirely understandable without reference to other sources. Please update your abstract.

We look forward to receiving your revised manuscript.

Kind regards,

Cathy W. S. Chen, Ph.D.

Academic Editor

PLOS ONE

Journal Requirements:

Additional Editor Comments (if provided):

The abstract should be self-contained and entirely understandable without reference to other sources. Please update your abstract.

Reviewers' comments:

Reviewer's Responses to Questions

**Comments to the Author**

1. If the authors have adequately addressed your comments raised in a previous round of review and you feel that this manuscript is now acceptable for publication, you may indicate that here to bypass the “Comments to the Author” section, enter your conflict of interest statement in the “Confidential to Editor” section, and submit your "Accept" recommendation.

Reviewer #1: (No Response)

Reviewer #2: All comments have been addressed

2. Is the manuscript technically sound, and do the data support the conclusions?

Reviewer #1: (No Response)

Reviewer #2: Yes

3. Has the statistical analysis been performed appropriately and rigorously? 

Reviewer #1: (No Response)

Reviewer #2: Yes

4. Have the authors made all data underlying the findings in their manuscript fully available?

Reviewer #1: (No Response)

Reviewer #2: Yes

5. Is the manuscript presented in an intelligible fashion and written in standard English?

Reviewer #1: (No Response)

Reviewer #2: Yes

6. Review Comments to the Author

Reviewer #1: (No Response)

Reviewer #2: I have read the revised manuscript entitled "An integer GARCH model for a Poisson process with time-varying zero-inflation". The authors followed my suggestions while revising their paper, so I am satisfied with the current version of the manuscript.

7. PLOS authors have the option to publish the peer review history of their article (what does this mean?). If published, this will include your full peer review and any attached files.

Reviewer #1: No

Reviewer #2: No

---

## [Author Response · Author response to Decision Letter 1]

13 Apr 2023

Cathy W.S. Chen, Ph.D.,

Academic Editor,

PLOS ONE.

Dear Dr. Chen,

Thank you for the invitation to submit a revised version of our manuscript (PONE-D-22-26550R1). We wish to thank you and the reviewers for their valuable time and insightful comments. These have greatly helped us to improve the quality of our manuscript. We considered the points raised by the academic editor during the review process, and carefully addressed them as described below. 

We updated the abstract and the new version of the abstract is self-contained and entirely understandable without reference to other sources. 

We believe that this new version of the manuscript incorporates all main aspects pointed out by the editor and hope that it satisfies the journal requirements and meets the high standards for publication in PLOS ONE. 

I am looking forward to hearing from you at your earliest convenience.

Thank you,

Your Sincerely,

Isuru Ratnayake

---

## [Editor Report · Decision Letter 2]

16 Apr 2023

PONE-D-22-26550R2An integer GARCH model for a Poisson process with time-varying zero-inflationPLOS ONE

Dear Dr. Ratnayake,

Thank you for submitting your manuscript to PLOS ONE. After careful consideration, we feel that it has merit but does not fully meet PLOS ONE’s publication criteria as it currently stands. Therefore, we invite you to submit a revised version of the manuscript that addresses the points raised during the review process.

My previous comment is, "The abstract should be self-contained and entirely understandable without reference to other sources. Please update your abstract.". Unfortunately, the authors do not follow it.

The following sentences should be removed or rewritten without any references.

"The proposed model is a generalization of the zero  inflated Poisson Integer GARCH model proposed by Fukang Zhu in 2012, which in return is a generalization of the Integer GARCH (INGARCH) model introduced by Ferland, Latour, and Oraichi in 2006.

We look forward to receiving your revised manuscript.

Kind regards,

Cathy W. S. Chen, Ph.D.

Academic Editor

PLOS ONE

Journal Requirements:

Additional Editor Comments:

My previous comment is, "The abstract should be self-contained and entirely understandable without reference to other sources. Please update your abstract.". Unfortunately, the authors do not follow it.

The following sentences should be removed or rewritten without any references.

"The proposed model is a generalization of the zero inflated Poisson Integer GARCH model proposed by Fukang Zhu in 2012, which in return is a

generalization of the Integer GARCH (INGARCH) model introduced by Ferland, Latour, and Oraichi in 2006.
---

## [Author Response · Author response to Decision Letter 2]

22 Apr 2023

The authors wish to thank the academic editor and the reviewers for their insightful and constructive comments, which helped to greatly enhance the quality and breath of this paper.

---

## [Editor Report · Decision Letter 3]

2 May 2023

An integer GARCH model for a Poisson process with time-varying zero-inflation

PONE-D-22-26550R3

Dear Dr. Ratnayake,

We’re pleased to inform you that your manuscript has been judged scientifically suitable for publication and will be formally accepted for publication once it meets all outstanding technical requirements.

Kind regards,

Cathy W. S. Chen, Ph.D.

Academic Editor

PLOS ONE

Additional Editor Comments (optional):

This manuscript is recommended to be accepted for publication.
---

## [Editor Report · Acceptance letter]

8 May 2023

PONE-D-22-26550R3 

An integer GARCH model for a Poisson process with time-varying zero-inflation 

Dear Dr. Ratnayake:

I'm pleased to inform you that your manuscript has been deemed suitable for publication in PLOS ONE. Congratulations! Your manuscript is now with our production department. 

Kind regards, 

on behalf of

Prof. Cathy W. S. Chen 

Academic Editor

PLOS ONE